# scDFM: Distributional Flow Matching for Robust Single-Cell Perturbation Prediction

**Chenglei Yu**[1,2]* **Chuanrui Wang**[2]* **Bangyan Liao**[1,2] **& Tailin Wu**[2]†
[1]Zhejiang University
[2]Department of Artificial Intelligence, School of Engineering, Westlake University
`{yuchenglei, wangchuanrui, wutailin}@westlake.edu.cn`

## Abstract

A central goal in systems biology and drug discovery is to predict the transcriptional response of cells to perturbations. This task is challenging due to the noisy, sparse nature of single-cell measurements and the fact that perturbations often induce population-level shifts rather than changes in individual cells. Existing deep learning methods typically assume cell-level correspondences, limiting their ability to capture such global effects. We present **scDFM**, a generative framework based on conditional flow matching that models the full distribution of perturbed cells conditioned on control states. By incorporating an MMD objective, our method aligns perturbed and control populations beyond cell-level correspondences. To further improve robustness to sparsity and noise, we propose the Perturbation-Aware Differential Transformer architecture (PAD-Transformer), a backbone that leverages gene interaction graphs and differential attention to capture context-specific expression changes. **scDFM** outperforms prior methods across multiple genetic and drug perturbation benchmarks, excelling in both unseen and combinatorial settings. In the combinatorial setting, it reduces MSE by 19.6% over the strongest baseline. These results highlight the importance of distribution-level generative modeling for robust *in silico* perturbation prediction. The code is available at https://github.com/AI4Science-WestlakeU/scDFM.

## 1 Introduction

Accurate prediction of the transcriptomic response of cells to genetic or drug perturbations at single-cell resolution is a central challenge in functional genomics and drug discovery (Bunne et al., 2023; Lotfollahi et al., 2023). Understanding these responses not only reveals complex gene regulatory networks, but also accelerates the design of novel therapeutic strategies and allows personalized medicine (Qi et al., 2024; Viñas Torné et al., 2025). However, given the exponential growth in the number of potential gene or drug combinations, systematically screening all possible perturbation combinations by experimental means is practically infeasible (Roohani et al., 2024). As a result, the development of *in silico* models capable of accurately predicting cellular perturbation effects has become critical, and progress in this area remains a primary barrier to advancements in the field.

A fundamental challenge in modeling single-cell perturbation responses lies in the unpaired nature of the data. Due to the destructive nature of RNA sequencing, it is impossible to observe the same cell both before and after perturbation. This makes cell-level pairing and supervision impossible, and standard pointwise losses ill-suited. Most existing models thus focus narrowly on recovering the mean expression profile, ignoring higher-order statistics such as variance, skewness, or shifts in subpopulation proportions (Mejia et al., 2025; Yu et al., 2025; Chi et al., 2025). This turns out to be a serious limitation; for example, Ramakrishnan et al. (2025) shows that perturbations induce complex distributional changes beyond the mean that many current methods fail to capture. Moreover, benchmarks such as Systema (Viñas Torné et al., 2025) show that simply predicting the population mean can perform even better than many sophisticated models under standard metrics (Ahlmann-Eltze et al., 2025; Csendes et al., 2025).

---

*Equal contribution.    †Corresponding author.

Moreover, single-cell transcriptomic data present severe modeling difficulties due to their sparse, zero-inflated, and noisy nature. Technical dropout often yields missing values that do not reflect true biological absence (Dai et al., 2024), while batch effects and uneven sequencing depth distort gene-gene correlations (Zhou et al., 2025). Moreover, perturbation effects are nonlinear and context-dependent, making the modeling task substantially harder (Xing & Yau, 2025; Song et al., 2025).

As a result, models that treat genes as independent inputs or rely on shallow architectures struggle to generalize to new cell types or out-of-distribution perturbations. When the model backbone treats genes independently without explicit gene co-expression relationships, it tends to overfit noise rather than extract meaningful biological signal. Empirical work reports that Geneformer (Theodoris et al., 2023) and scGPT (Cui et al., 2024) underperform simpler baselines with standard batch correction tools in zero-shot settings (Kedzierska et al., 2025). This underscores the need for more expressive, perturbation-robust, and noise-resistant model backbones.

We propose **scDFM**, a generative framework based on conditional flow matching (Lipman et al., 2022), to accurately reconstruct the distribution of single-cell gene expression after perturbations. Our model tackles the two major limitations mentioned above: the neglect of population-level distributional fidelity and the failure to account for noisy, interdependent gene regulation.

First, to address distribution-level fidelity, we incorporate the Maximum Mean Discrepancy (MMD) loss (Gretton et al., 2006) into our training objective. Unlike traditional loss functions between paired prediction and ground truth, the MMD loss directly quantifies the distance between the cell distribution generated by the model and the true post-perturbation cell distribution. By combining the MMD loss with the flow matching objective, the model is guided to reproduce the overall population shift induced by perturbations rather than merely improving per-cell predictions, thereby better capturing distribution-level effects.

Second, to mitigate noise and capture complex regulatory dependencies, we introduce the Perturbation-Aware Differential Transformer (PAD-Transformer). It incorporates gene–gene masked attention, where a co-expression graph guides the model to focus on biologically related genes, and employs differential attention to separate control and perturbation signals and highlight their interactions. By integrating these components into a unified backbone, PAD-Transformer filters noise, preserves regulatory structure, and scales effectively to complex perturbation scenarios.

We evaluate our method on two challenging benchmarks: (1) the Norman combinatorial gene-perturbation dataset (Norman et al., 2019), and (2) the Combosiplex combinatorial drug-perturbation dataset (Mathur et al., 2022). On the Norman dataset, we assess generalization under two settings: (i) an additive setting, where all singles and a subset of duals are used for training, and (ii) a hold-out setting, where specific dual combinations are completely excluded from training. Together, these evaluations demonstrate that the synergy between biologically structured attention and distribution-level training in **scDFM** significantly improves both accuracy and robustness of in silico single-cell perturbation prediction.

## 2 RELATED WORK

**Foundation Models for Single-Cell Biology.** Recent advances in large-scale pretraining have led to the development of foundation-style models for single-cell data, such as Geneformer (Theodoris et al., 2023), scBERT (Yang et al., 2022), scFoundation (Khan et al., 2023), scGPT (Cui et al., 2024), and UCE (Rosen et al., 2023). These models are typically trained on large collections of unperturbed expression profiles to learn general-purpose embeddings of cells and genes, which can then be transferred to downstream tasks with minimal supervision. While such approaches enable broad applicability and data-efficient learning, several studies (Ahlmann-Eltze et al., 2025; Csendes et al., 2025; Kedzierska et al., 2025) have shown that they may struggle to capture perturbation-specific effects, especially when distributional changes go beyond the population mean.

**Models Tailored to Perturbation Effects.** A parallel line of work has proposed perturbation-specific models that more directly capture gene regulatory dynamics. CPA (Lotfollahi et al., 2023) models the compositional structure of perturbations by embedding genes and conditions in a shared latent space, enabling extrapolation to novel combinations. GEARS (Roohani et al., 2024) incorporates biological priors such as gene–gene co-expression and Gene Ontology into the model archi-

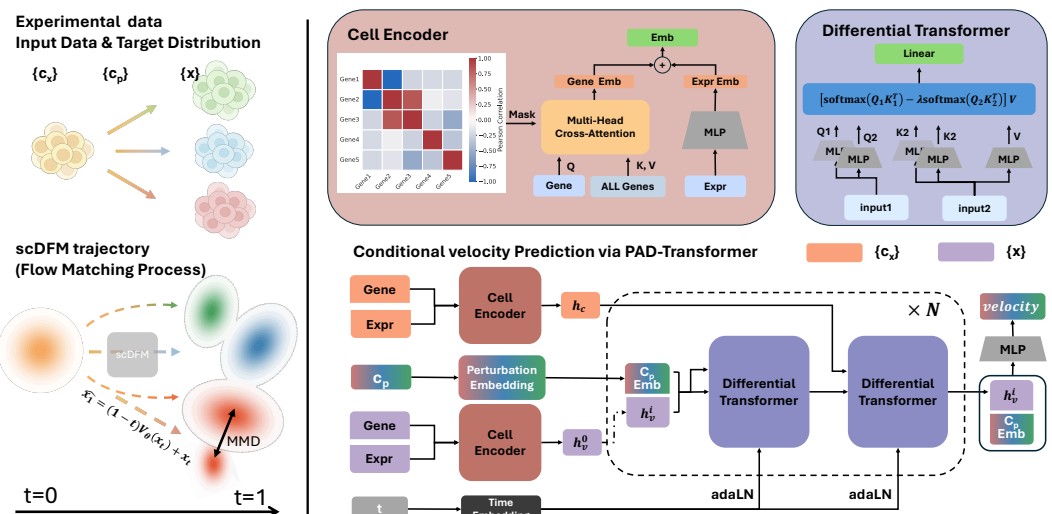

Figure 1: Overview of scDFM, which models perturbation-specific cell state transitions as a flow matching process from noise to perturbed expression. The PAD-Transformer predicts time-dependent velocities conditioned on control cell context and perturbation embedding, while gene–gene masked attention and differential Transformer layers capture biological dependencies. Final distributional alignment is enforced via MMD regularization.

tecture, improving generalization to unseen perturbations. PerturbNet (Yu et al., 2025) and GPerturb (Xing & Yau, 2025) further explore probabilistic formulations for unseen condition prediction.

More recently, generative approaches such as scDiffusion (Bunne et al., 2023), CellFlow (Klein et al., 2025), and UNLASTING (Chi et al., 2025) have applied diffusion-based models to learn continuous trajectories of cellular transitions under perturbation. These models are often framed in latent space and provide a principled way to interpolate between control and perturbed states. However, their performance is sensitive to the choice of embedding space. Our proposed method draws connections to both lines of prior work. Like foundation models, we leverage attention-based encoders to extract rich representations from unperturbed expression profiles. At the same time, our method explicitly models the perturbation-induced transition dynamics in the expression space with flow matching (Lipman et al., 2022).

## 3 METHOD

**Problem Setup.** Let $\mathcal{G} = \{g_1, g_2, \ldots, g_G\}$ denote the set of $G$ profiled genes. The pre-perturbation state of a cell is represented as $c_x = (c_x(g_1), \ldots, c_x(g_G)) \in \mathbb{R}_+^G$, where $c_x(g_i)$ is the expression level of gene $g_i$ in the unperturbed cell. Analogously, the post-perturbation state is $x = (x(g_1), \ldots, x(g_G)) \in \mathbb{R}_+^G$, where $x(g_i)$ denotes the expression level of gene $g_i$ after perturbation. The perturbation condition is encoded as a multi-hot vector $c_p \in \{0, 1\}^d$, with $c_p[j] = 1$ if the $j$-th perturbation (*e.g.*, a drug or CRISPR guide) is applied. Formally, each training instance consists of $(c_x, c_p, x)$, and the goal is to learn the conditional generative model $p_\theta(x \mid c_x, c_p)$ that captures the population-level distribution of perturbed cell states and generalizes to novel perturbation combinations not observed during training.

**Model Overview.** Figure 1 provides an overview of **scDFM**, which is built on a flow matching architecture (Section 3.1). The model learns continuous trajectories that transform noisy initial states into target perturbed expressions through iterative conditional refinement, conditioned on the control expression $c_x$ and the perturbation signal $c_p$. At each step, gene expression features are encoded with a gene–gene correlation mask (Section 3.3), while perturbation and time embeddings are injected into stacked PAD-Transformer blocks to capture perturbation-aware dynamics (Section 3.4). To ensure fidelity at both local and global levels, training combines the conditional flow matching loss

with a multi-kernel MMD regularizer (Section 3.2), while a velocity head estimates instantaneous changes along the trajectory.

## 3.1 Perturbation Prediction with Flow Matching

Flow Matching (FM) (Lipman et al., 2022) is a continuous-time generative modeling framework that learns a time-dependent velocity field to morph a source distribution into a target distribution along a continuous trajectory. In this work, we make the first attempt to apply the FM framework directly in the high-dimensional gene expression space. Specifically, the source distribution $x_0$ is defined as the noisy gene expression distribution, and the target distribution $x_1$ is defined as the perturbed gene expression distribution. The transformation evolves over a denoising time interval $t \in [0, 1]$, conditioned on both the control state $c_x$ (the pre-perturbation gene expression) and the perturbation condition $c_p$ (which may correspond to single or combinatorial perturbations).

Formally, our objective is to learn a conditional velocity field $v_\theta(x_t \mid t, c_x, c_p)$. This field characterizes the instantaneous rate of change of the cell state at denoising time $t$. The state evolution follows the conditional ODE:

$$\frac{dX_t}{dt} = v_\theta(X_t \mid t, c_x, c_p), \tag{1}$$

where $X_t$ denotes the generated gene expression state at time $t$.

During training, Conditional Flow Matching (CFM) minimizes the discrepancy between the predicted velocity $v_\theta$ and the reference velocity $v$ induced by a predefined interpolation path $\pi_t(x_0, x_1)$, for which we adopt the linear form $\pi_t(x_0, x_1) = (1-t)x_0 + tx_1$. Given a control state $c_x$ which tells us cell line identity and a perturbation condition $c_p$, we define $x_0 \sim q_0$ as a noisy source expression, and $x_1 \sim q_1(\cdot \mid c_x, c_p)$ as the corresponding post-perturbation expression drawn from the target distribution. The training objective is given by:

$$\mathcal{L}_{\text{CFM}}(\theta) = \mathbb{E}_{c_x, c_p} \, \mathbb{E}_{x_0 \sim q_0, x_1 \sim q_1(\cdot | c_x, c_p)} \, \mathbb{E}_{t \sim \mathcal{U}(0,1)} \left[ \left\| v_\theta(x_t \mid t, c_x, c_p) - v(x_t \mid x_0, x_1, t, c_x, c_p) \right\|_2^2 \right]. \tag{2}$$

This formulation enables the model to directly learn the conditional transformation from noisy intermediate states to the true post-perturbation states, while explicitly incorporating both the initial control state and the perturbation condition.

## 3.2 Flow Matching with Multi-Kernel MMD Regularization

Our framework learns a conditional flow matching (FM) process by optimizing a velocity field $v_\theta$ that aligns the generated trajectories with reference perturbation dynamics. This encourages biologically plausible and coherent evolution over continuous time. However, FM alone enforces local dynamical consistency and does not guarantee that the terminal distribution of generated cells $\hat{X}$ statistically aligns with the ground-truth perturbed distribution $X$.

To address this limitation and ensure population-level fidelity, we introduce a distribution-level regularization term based on the *Maximum Mean Discrepancy* (MMD), which directly compares the predicted terminal distribution to the empirical distribution of ground-truth perturbed cells. We choose MMD over KL divergence or Wasserstein distance because it is directly sample-based, computationally efficient, and robust under support mismatch, making it well-suited for high-dimensional single-cell settings.

**One-step prediction and target endpoint distribution.** At each training step, for a sampled intermediate state $x_t \sim \pi_t(x_0, x_1|c_x, c_p)$, we apply the learned velocity field $v_\theta(x_t|t, c_x, c_p)$ to compute a one-step prediction of the perturbed state:

$$\hat{x}_1 = x_t + (1-t) \cdot v_\theta(x_t|t, c_x, c_p), \qquad \text{with } c_x \sim D_{\text{Control}}$$

where $D_{\text{Control}}$ denotes the empirical distribution of pre-perturbation (control) cells. This formulation approximates the endpoint of the flow. This yields a batch of model-predicted terminal samples $\{\hat{x}_1^{(i)}\} \sim \hat{X}_1, i \in [0, B]$, which we compare against empirical samples $\{x_1^{(i)}\} \sim X_1$ drawn from the ground-truth post-perturbation cell population.

**Multi-kernel MMD regularizer.** To measure the discrepancy between $\hat{X}_1$ and $X_1$, we use the squared MMD with a mixture of Gaussian RBF kernels:

$$k_{\mathrm{mix}}(x, x') \;=\; \frac{1}{L} \sum_{\ell=1}^{L} \exp\Big( - \frac{\|x-x'\|^2}{2\sigma_\ell^2} \Big), \tag{3}$$

where $\{\sigma_\ell\}_{\ell=1}^{L}$ are bandwidths selected via a median heuristic. In practice, we estimate a reference scale from pairwise squared distances and generate a small set of bandwidths by multiplying this scale with fixed factors, which stabilizes training across heterogeneous cell populations. While Conditional Flow Matching (CFM) provides a principled way to learn velocity fields that interpolate between control and perturbed states, it primarily enforces local dynamical consistency along trajectories. This means that the model is trained to match instantaneous velocity fields but does not directly constrain the global distributional outcome. As a result, the terminal distribution of generated cells $\hat{X}$ may deviate from the ground-truth perturbed distribution $X$, leading to mismatches in population-level statistics or biologically relevant gene expression patterns.

The final training objective combines pointwise loss and distribution-level signals:

$$\mathcal{L} \;=\; \mathcal{L}_{\mathrm{CFM}} \;+\; \lambda\,\mathcal{L}_{\mathrm{MMD}}, \tag{4}$$

$$\mathcal{L}_{\mathrm{MMD}}(\theta) = \mathbb{E}_{c_x, c_p}\, \mathbb{E}_{x_0 \sim q_0, x_1 \sim q_1(\cdot|c_x, c_p)}\, \mathbb{E}_{t \sim \mathcal{U}(0,1)}\, k_{\mathrm{mix}}\big(x_1, \hat{x}_1(x_t, t, c_x, c_p)\big) \tag{5}$$

where $\lambda > 0$ balances trajectory consistency against endpoint distributional fidelity. This combination allows the model to learn both the fine-grained trajectory of individual cells and the global shift in the cell population distribution, addressing both local and global fidelity. The specific configuration and algorithm of the MMD regularization is provided in Appendix A.2.

### 3.3 Initial Embedding of Gene Expressions

**Gene Encoding.** Given a control cell expression profile $c_x \in \mathbb{R}^G$, a perturbation condition $c_p$, the current timestep $t$, and the corresponding perturbed expression $x_t \in \mathbb{R}^G$, we construct cell representations over a selected subset of genes $\mathcal{S} \subseteq \mathcal{G}$:

$$h_c = E_v\big(c_x^{(\mathcal{S})}\big) + E_g(\mathcal{S}), \quad h_t^0 = E_v\big(x_t^{(\mathcal{S})}\big) + E_g(\mathcal{S}), \tag{6}$$

where $c_x^{(\mathcal{S})}$ and $x_t^{(\mathcal{S})}$ denote the expression values restricted to the selected gene subset $\mathcal{S}$. Here $E_v$ maps each expression value to a $d$-dimensional embedding, while $E_g(\mathcal{S}) \in \mathbb{R}^{|\mathcal{S}| \times d}$ is a sequence of contextualized gene embeddings obtained from a cross-attention based gene encoder. The cross-attention mask is derived from gene-gene relationships within the dataset, ensuring that each gene token interacts only with biologically relevant neighbors.

**Gene-gene co-expression graph as attention mask.** Relying solely on the cross-attention network is insufficient to capture the intrinsic dependencies among genes. In reality, genes are organized within complex regulatory networks, which directly determine the transcriptomic changes under perturbations. Ignoring such dependencies may cause the model to treat each gene as an independent feature, thereby limiting the biological plausibility of the predictions.

To address this issue, we construct a *gene–gene co-expression graph* from the training data and incorporate it into the attention mechanism. For any two genes $i$ and $j$, we define the edge weight as the absolute Pearson correlation:

$$w_{ij} = \left| \frac{\mathrm{Cov}(x_i, x_j)}{\sigma(x_i), \sigma(x_j)} \right|, \tag{7}$$

where $x_i$ and $x_j$ are the expression vectors of genes $i$ and $j$ across all cells. Based on the weight matrix $W = (w_{ij})$, we apply a KNN strategy to select the most strongly correlated (positive or negative) neighbors for each gene, yielding a sparse adjacency matrix $\tilde{A}$.

This static graph serves as a biologically grounded prior and is used to construct a sparse attention mask in the gene encoder $E_g$, thereby constraining the attention mechanism to focus on biologically meaningful interactions.

### 3.4 BACKBONE DESIGN

A core challenge in modeling single-cell perturbation responses lies in the noisy, sparse, and high-dimensional nature of the data. To address this, we introduce the Perturbation-Aware Differential Transformer (PAD-Transformer), a backbone that injects perturbation signals at every layer while employing differential attention to suppress spurious correlations from noisy genes. By jointly encoding control and perturbed states, the model captures explicit dependencies between them. Its architecture combines a **differential attention module** with a **perturbation-aware latent refinement block**, enabling robust modeling of perturbation-specific dynamics under challenging single-cell conditions.

**Differential Attention Module.** Standard Transformers are often prone to over-attending to irrelevant tokens, especially in noisy biological data. This is particularly problematic in perturbation modeling, where only a subset of genes may respond while others should remain suppressed. To address this, we incorporate a differential attention mechanism (Ye et al., 2025), which computes attention as the difference between two softmax distributions:

$$A_1 = \text{softmax}\left(\frac{Q_1 K_1^\top}{\sqrt{d_h}}\right), \quad A_2 = \text{softmax}\left(\frac{Q_2 K_2^\top}{\sqrt{d_h}}\right), \tag{8}$$

$$\alpha_{\text{diff}} = A_1 - \lambda A_2, \quad \text{DiffAttn}(X, Y) = \sum \alpha_{\text{diff}}^i V^i, \tag{9}$$

where $\lambda$ is a learnable scaling factor, $Q_i = W_{Q_i} X$, $K_i = W_{K_i} Y$, and $V = W_V Y$.

**Latent Refinement.** At each layer, PAD-Transformer refines the latent representation $h_v^\ell$ via three operations:

1. **Perturbation injection.** The perturbation condition $c_p$ is embedded as $e_p$, broadcast, concatenated with $h_v^\ell$, and passed through an MLP adapter to obtain the injected representation:

$$\bar{h}_v^\ell = \text{MLP}_\ell\Big(\big[\, h_v^\ell \ \|\ \mathbf{1}_T \otimes e_p \,\big]\Big). \tag{10}$$

2. **Self-differential attention.** Applied to $\bar{h}_v^\ell$ to suppress noisy activations and refine informative variations within the latent representation:

$$\tilde{h}_v^\ell = \bar{h}_v^\ell + \text{DiffAttn}(X = \bar{h}_v^\ell, Y = \bar{h}_v^\ell; t_{\text{emb}}). \tag{11}$$

3. **Cross-differential attention.** Incorporates the control representation $h_c$ as a reference to guide refinement of the perturbed latent:

$$h_v^{\ell+1} = \tilde{h}_v^\ell + \text{DiffAttn}(X = \tilde{h}_v^\ell, Y = h_c; t_{\text{emb}}). \tag{12}$$

The timestep $t$ is encoded as $t_{\text{emb}} = \text{MLP}(\text{SinCos}(t))$, where $\text{SinCos}(t)$ denotes a sinusoidal embedding of $t$ at multiple frequencies. This embedding provides adaLN-Zero modulation (Peebles & Xie, 2023) for every self-differential attention and cross-differential attention layer.

**Output.** After $L$ layers, $e_p$ is concatenated again and the decoder produces the predicted perturbed state:

$$\hat{x} = D([\, h_v^L \| \mathbf{1}_T \otimes e_p \,]). \tag{13}$$

PAD-Transformer leverages perturbation-aware differential attention to refine latent trajectories, ensuring robust modeling of both cell-level dynamics and population-level transcriptional shifts. The complete algorithmic workflow and training procedure are provided in Appendix A.3.

## 4 EXPERIMENT

### 4.1 EXPERIMENTAL SETUP

**Baselines.** We benchmark our method against a broad set of baselines spanning simple statistical models to deep learning and foundation-model approaches. These include autoencoder-based

model CPA (Lotfollahi et al., 2023), graph-based model GEARs (Roohani et al., 2024), two single-cell foundation models Geneformer (Theodoris et al., 2023) and scGPT (Cui et al., 2024), as well as a naive linear regression baseline (Ahlmann-Eltze et al., 2025), and statistical mean expression over control cells. In addition, we include CellFlow (Klein et al., 2025), a flow-matching model trained in a reduced 50-dimensional PCA space and conditioned on perturbation identity, as well as State (Adduri et al., 2025), a one-step perturbation response model that conditions on basal and perturbation embeddings. All models are trained using the log of expression values for all genes, but at evaluation time the prediction target is restricted to the top 1,000 highly variable genes. Geneformer is evaluated with pretrained weights, whereas scGPT and STATE are trained from scratch without pretraining (Appendix A.5).

**Metrics.** We evaluate model performance from three complementary perspectives. (1) At the whole-transcriptome level, MAE, MSE, Pearson $\Delta$ and mean L2 errors quantify pointwise reconstruction accuracy across all genes. We additionally include Pearson $\hat{\Delta}$ and $\hat{\Delta}_{20}$, computed using the training-perturbation mean as the reference. These variants mitigate control–perturbation baseline bias and follow the updated evaluation practice introduced by Systema (Viñas Torné et al., 2025). (2) At the distribution level, the Discrimination Score (DS) (Roohani et al., 2025) assesses whether predicted populations under different perturbations remain well separated, penalizing collapsed or averaged responses. (3) At the differential expression (DE) level, DE-Spearman $\rho$ quantifies the rank correlation between predicted and real log fold changes, computed only on the set of statistically significant DE genes. This metric evaluates whether the model correctly captures both the directionality and the relative ordering of differential expression signals (details in Appendix A.4.4).

**Data.** We evaluate our method on two single-cell perturbation datasets: Norman and ComboSciPlex. The Norman dataset (Norman et al., 2019) consists of genetic perturbations (CRISPR-based overexpression) in the K562 cell line, including both single and double perturbations (Appendix A.4.1). The ComboSciPlex dataset involves drug perturbations, which differ in modality from the genetic perturbations in Norman, thus allowing assessment of cross-domain generalization (Srivatsan et al., 2020; Lotfollahi et al., 2023) (Appendix A.4.2). To ensure robust evaluation, we perform multiple random re-splits and report averaged results for Norman dataset. Finally, we include ablation analysis to showcase the effect of each component of our model.

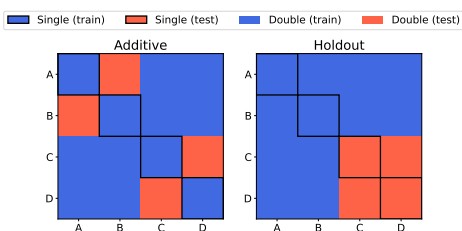

Figure 2: The **additive** setting (Section 4.2) tests generalization to unseen doubles when all singles are observed, while the **holdout** setting (Section 4.3) evaluates prediction of entirely unobserved singles and their combinations

We apply two evaluation splits: an additive setting, where all single perturbations used in combinations for testing have been observed individually in training; and a holdout setting, where certain single perturbations and all combinations involving them together are entirely withheld during training. These two settings together allow us to assess performance both in additive and more challenging generalization (holdout) regimes.

## 4.2 EVALUATING ON SEEN SINGLE PERTURBATIONS (ADDITIVE SETTING)

In the first experiment, we use the additive split, where test combinations are composed of single perturbations already seen during training (Fig. 2 left). Table 1 shows that our model achieves the best or near-best results across all categories: at the global reconstruction level it yields the lowest L2, MSE, and MAE; at the distribution level it achieves the highest discrimination score with a more compact error distribution (Fig. 3); and at the DE level it obtains the strongest DE-Spearman $\rho$. Notably, the Additive baseline itself is competitive, consistent with recent findings (Ahlmann-Eltze et al., 2025).

By contrast, existing methods show instability. scGPT drops sharply in DE-Spearman $\rho$, CPA suffers from large pointwise errors, and even Geneformer and GEARS trade off lower L2 for weaker

distributional alignment. In comparison, our approach maintains a balanced profile across metrics, achieving robustness without sacrificing one dimension of performance for another. We further include a case study on the CEBPE+CEBPA perturbation prediction in Appendix A.7.

Quantitatively, our model reduces the MSE by 19.6% compared to CellFlow (0.00315 vs. 0.00392), the second best performing model, and obtains a lower MAE (0.02155 vs. 0.02207), while also achieving the highest discrimination score (0.9737). Under the stricter Systema-style metrics, it further attains the strongest Pearson $\hat{\Delta}_{20}$ (0.9260), indicating that the improvements persist even when evaluation penalizes expression-overlap effects.

Table 1: Comparison of different methods across evaluation metrics on the Norman additive split.

| Model | L2 ↓ | MSE ↓ | MAE ↓ | DE-Spearman $\rho$ ↑ | Pearson $\Delta$ ↑ | DS ↑ | Pearson $\hat{\Delta}$ ↑ | Pearson $\hat{\Delta}_{20}$ ↑ |
|---|---|---|---|---|---|---|---|---|
| Control | 3.9937 | 0.01839 | 0.03953 | N.A. | N.A. | 0.5135 | -0.1695 | -0.1297 |
| Additive | 1.9395 | 0.00448 | 0.02276 | 0.5564 | **0.9024** | 0.9686 | **0.8584** | 0.9244 |
| scGPT | 3.4112 | 0.01349 | 0.03796 | 1.07e-5 | 0.5304 | 0.5404 | 0.2165 | 0.2414 |
| Geneformer | 1.9132 | 0.00410 | 0.02360 | 0.3741 | 0.7732 | 0.8241 | -0.0078 | 0.2239 |
| GEARS | 3.5531 | 0.01387 | 0.06624 | 0.5624 | 0.7421 | 0.8601 | -0.0089 | 0.2032 |
| CPA | 5.7629 | 0.03435 | 0.07894 | 0.0713 | 0.3845 | 0.6021 | -0.0039 | 0.2254 |
| STATE | 17.3330 | 0.30059 | 0.24705 | 0.5288 | -0.0108 | 0.5135 | -0.0069 | 0.2515 |
| CellFlow | 1.7064 | 0.00392 | 0.02207 | 0.5503 | 0.8678 | 0.9321 | 0.8395 | 0.8988 |
| **scDFM** (ours) | **1.7043** | **0.00315** | **0.02155** | **0.5705** | 0.8853 | **0.9737** | 0.8468 | **0.9260** |

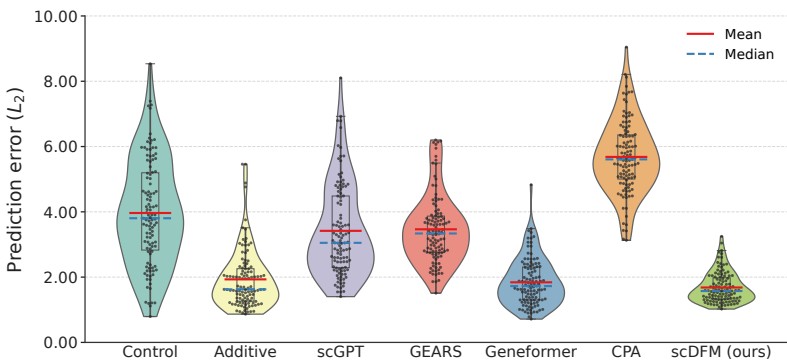

Figure 3: Double perturbation prediction error ($L_2$). Our method achieves the lowest error distribution, outperforming both additive and baseline models.

## 4.3 EVALUATING ON UNSEEN PERTURBATIONS (HOLDOUT SETTING)

In this experiment, we adopt a holdout split to test how well our model generalizes to perturbations not seen alone during training. Specifically, we remove a subset of single perturbations along with all double perturbations involving them from the training set, and use these held-out conditions for testing (Fig. 2 right). This allows us to assess generalization both for unseen individual perturbations and for their combinatorial effects.

Since this task poses greater challenges, we observe larger performance gaps across models. As shown in Tab. 2, each baseline exhibits distinct limitations: **scGPT** fails to capture perturbation direction, resulting in negative DE-Spearman $\rho$; **GEARS** improves correlation metrics but suffers from large pointwise errors; **Geneformer** achieves low L2 and MAE yet falls short in preserving distributional structure. In contrast, our method combines structural priors and MMD regularization to achieve both low error and strong distributional fidelity, ensuring robust generalization.

## 4.4 EVALUATING DRUG PERTURBATIONS

Table 3 reports performance on the ComboSciPlex dataset (Srivatsan et al., 2020), which measures drug rather than genetic perturbations. At the global reconstruction level, our model achieves the lowest L2, MSE, and MAE, indicating the most accurate transcriptome recovery. At the distribution level, it maintains competitive discrimination scores, slightly below CPA but more stable than

Table 2: Comparison of different methods across evaluation metrics on the Norman holdout split.

| Setting | Model | L2 ↓ | MSE ↓ | MAE ↓ | DE-Spearman $\rho$ ↑ | Pearson $\Delta$ ↑ | DS ↑ | Pearson $\hat{\Delta}$ ↑ | Pearson $\hat{\Delta}_{20}$ ↑ |
|---|---|---|---|---|---|---|---|---|---|
| Single | Control | 2.6834 | 0.0095 | 0.0263 | N.A. | N.A. | 0.5217 | 0.1618 | 0.1982 |
| | scGPT | 2.5007 | 0.0080 | 0.0259 | -0.1139 | 0.4503 | 0.5680 | 0.0747 | 0.0798 |
| | GEARS | 2.5641 | 0.0075 | 0.0466 | 0.3569 | 0.6646 | 0.8271 | 0.6356 | 0.7914 |
| | Geneformer | 1.6962 | 0.0036 | 0.0191 | 0.3669 | 0.6955 | 0.8070 | 0.5620 | 0.6513 |
| | CPA | 5.8060 | 0.0356 | 0.0853 | 0.1168 | 0.2837 | 0.5796 | -0.0028 | 0.0802 |
| | STATE | 18.2543 | 0.3333 | 0.2693 | 0.6116 | 0.0004 | 0.5236 | 0.0154 | 0.2386 |
| | CellFlow | 1.6758 | 0.0035 | 0.0191 | 0.2860 | 0.7109 | 0.8072 | 0.6138 | 0.6753 |
| | **scDFM** (ours) | **1.6186** | **0.0030** | **0.0190** | **0.6957** | **0.7127** | **0.8914** | **0.6659** | **0.8116** |
| Double | Control | 4.1882 | 0.0207 | 0.0423 | N.A. | N.A. | 0.5322 | -0.1303 | -0.0265 |
| | scGPT | 3.5171 | 0.0153 | 0.0362 | -0.0665 | 0.5693 | 0.5578 | 0.2814 | 0.2652 |
| | GEARS | 3.7458 | 0.0156 | 0.0708 | 0.2543 | 0.7552 | 0.8766 | 0.6407 | 0.8413 |
| | Geneformer | 2.0819 | 0.0050 | 0.0237 | 0.3468 | 0.7361 | 0.8067 | 0.6245 | 0.7261 |
| | CPA | 5.7891 | 0.0357 | 0.0796 | 0.3652 | 0.4176 | 0.6311 | 0.2432 | 0.2870 |
| | STATE | 18.4458 | 0.3404 | 0.2733 | 0.4071 | 0.0061 | 0.5289 | -0.0023 | 0.2580 |
| | CellFlow | 2.1042 | 0.0049 | 0.0236 | 0.5074 | 0.8095 | 0.8622 | 0.6780 | 0.7155 |
| | **scDFM** (ours) | 2.0309 | **0.0047** | **0.0235** | **0.5676** | **0.8357** | **0.9189** | **0.7769** | **0.8688** |

scGPT. At the DE level, it achieves the highest Pearson $\Delta$ and DE-Spearman $\rho$, demonstrating superior recovery of differential expression patterns. Overall, these results show that our approach generalizes effectively to drug perturbations, combining low pointwise error with consistent capture of perturbation-specific signals.

Table 3: Comparison of different methods across evaluation metrics on Combosciplex.

| Model | L2 ↓ | MSE ↓ | MAE ↓ | DE-Spearman $\rho$ ↑ | Pearson $\Delta$ ↑ | DS ↑ |
|---|---|---|---|---|---|---|
| Control | 5.3716 | 0.0324 | 0.0698 | N.A. | N.A. | 0.5714 |
| scGPT | 1.6934 | 0.0031 | 0.0251 | -0.1261 | 0.8322 | 0.8571 |
| CPA | 1.6592 | 0.0029 | 0.0240 | 0.7906 | 0.8150 | **0.8980** |
| **scDFM** (ours) | **1.6567** | **0.0028** | **0.0220** | **0.8289** | **0.8933** | 0.8776 |

## 4.5 ABLATION STUDY

To assess the contribution of each component, we perform ablation experiments on the Norman dataset under the holdout split. Figure 5 reports quantitative results. Removing the gene-gene mask or the Differential Transformer backbone reduces correlation with ground truth, highlighting the importance of structural priors and noise suppression. Dropping MMD regularization causes the sharpest decline, underscoring its critical role in distribution-level fidelity.

In addition to quantitative metrics, we also visualize representative perturbations in Fig. 4. Without MMD, generated cells deviate substantially from the ground truth distribution. In contrast, our full model preserves the global geometry and population structure, demonstrating that MMD is essential for stable and biologically consistent predictions.

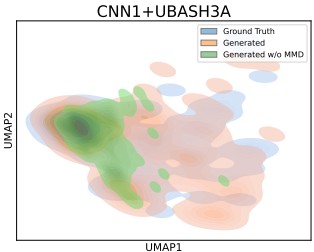 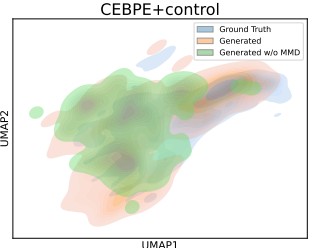 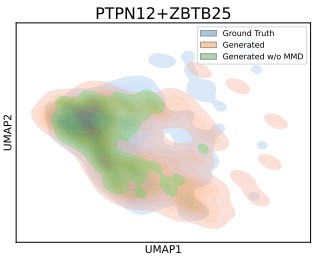

Figure 4: UMAP visualizations of perturbed cell states. Removing MMD leads to clear distributional mismatches, where generated cells deviate from the ground truth manifold.

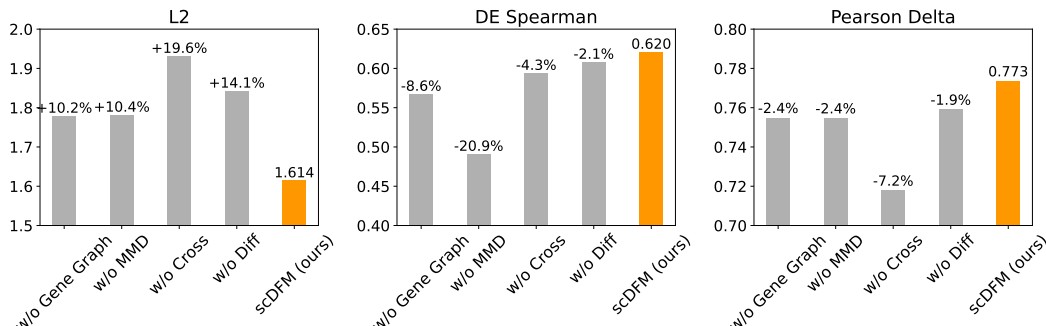

Figure 5: Ablation study on the Norman holdout setting.

## 5 CONCLUSION

In this paper, we have presented `scDFM`, a distribution-aware flow matching framework for robust single-cell perturbation prediction. By integrating conditional flow matching with MMD-based alignment and a perturbation-aware differential Transformer, our method captures both local dynamics and global population shifts. Extensive evaluations across genetic and drug perturbations demonstrate that `scDFM` achieves strong generalization to unseen combinations while maintaining low error and high biological fidelity.

Overall, our work paves the way for distribution-aware generative models that can serve as digital twins of cellular responses, providing a foundation for in silico screening and systems-level biology. Limitations and directions for future work are discussed in Appendix A.6.

### ACKNOWLEDGEMENT

We thank Minsi Ren and Tian Xia for helpful discussions and valuable feedback on the manuscript. We also gratefully acknowledge the support from the Westlake University Research Center for Industries of the Future and the Westlake University Center for High-Performance Computing. The content of this work is solely the responsibility of the authors and does not necessarily represent the official views of the funding entities.

### ETHICS STATEMENT

This study exclusively uses publicly available single-cell gene expression datasets, in accordance with their respective data use agreements. No personally identifiable or patient-specific information was used in this work. The methods developed in this study are intended for in silico modeling and hypothesis generation in biological research. While our approach could potentially assist drug discovery or therapeutic design in the future, it is not intended for direct clinical use. We encourage responsible and transparent use of generative models in biology, and caution against over-interpretation of model predictions without rigorous experimental validation.

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

CONTENTS

# A  APPENDIX

## A.1  THE USE OF LARGE LANGUAGE MODELS (LLMS)

In this paper, we employ LLMs as general-purpose assist tools for text refinement and language polishing. All core research ideas, datasets, and scientific conclusions presented in this paper are our own original contributions.

## A.2  MMD DETAILS

**Dynamic kernel selection for MMD.**  We implement MMD with a multi-kernel Gaussian RBF mixture:

$$k_{\text{mix}}(x, x') = \frac{1}{L} \sum_{\ell=1}^{L} \exp\Big( - \frac{\|x - x'\|^2}{2\sigma_\ell^2} \Big).$$

Rather than fixing the kernel bandwidths $\{\sigma_\ell\}$, we dynamically adjust them at each training step. Specifically, we compute pairwise squared distances within the batch:

$$D_{ij} = \|x_i - x_j\|^2,$$

take the median of all off-diagonal entries as a reference scale $m$, and generate multiple bandwidths as

$$\sigma_\ell = \sqrt{s_\ell \cdot m}, \quad s_\ell \in \{0.5, 1.0, 2.0, 4.0\}.$$

This procedure ensures that kernel bandwidths adapt to the distributional scale of the current mini-batch, while the use of multiple scales improves robustness to heterogeneous gene expression ranges.

The unbiased squared MMD estimate is then computed as

$$\text{MMD}^2(X, Y) = \frac{1}{m(m-1)} \sum_{i \neq j} k_{\text{mix}}(x_i, x_j) + \frac{1}{n(n-1)} \sum_{i \neq j} k_{\text{mix}}(y_i, y_j) - \frac{2}{mn} \sum_{i,j} k_{\text{mix}}(x_i, y_j).$$

---

**Algorithm 1:** Dynamic multi-kernel RBF selection and unbiased MMD$^2$

---

**Input:** Real samples $X = \{x_i\}_{i=1}^m$, generated samples $Y = \{y_j\}_{j=1}^n$, factors $\mathcal{S} = \{0.5, 1.0, 2.0, 4.0\}$
**Output:** $\text{MMD}^2(X, Y)$

1 **Pairwise distances:** $D_{xx}[i,j] \leftarrow \|x_i - x_j\|^2$, $D_{yy}[i,j] \leftarrow \|y_i - y_j\|^2$, $D_{xy}[i,j] \leftarrow \|x_i - y_j\|^2$.;
2 **Batch-adaptive bandwidths (median heuristic):** $m \leftarrow \max\big(\text{median}\{D_{xx}[i,j] : i \neq j\}, \varepsilon\big)$;;
3 $\Sigma \leftarrow \{\sigma = \sqrt{s\,m} \mid s \in \mathcal{S}\}$.;
4 **for** $\sigma \in \Sigma$ **do**
5      $\beta \leftarrow \frac{1}{2\sigma^2 + \varepsilon}$;;
6      $K_{xx} \leftarrow e^{-\beta D_{xx}}$, $K_{yy} \leftarrow e^{-\beta D_{yy}}$, $K_{xy} \leftarrow e^{-\beta D_{xy}}$;;
7      $u_{xx} \leftarrow \dfrac{\sum K_{xx} - \sum \text{diag}(K_{xx})}{m(m-1) + \varepsilon}$;;
8      $u_{yy} \leftarrow \dfrac{\sum K_{yy} - \sum \text{diag}(K_{yy})}{n(n-1) + \varepsilon}$;;
9      $u_{xy} \leftarrow \dfrac{\sum K_{xy}}{mn}$;;
10      $v(\sigma) \leftarrow u_{xx} + u_{yy} - 2u_{xy}$.;
11 **return** $\text{MMD}^2(X, Y) = \dfrac{1}{|\Sigma|} \sum_{\sigma \in \Sigma} v(\sigma)$.;

---

## A.3  TRAINING AND INFERENCE DETAILS

**Setup.**  Let the dataset be $\mathcal{D} = \{(c_x, x_1, c_p)\}$ measured over $G$ genes with a gene–gene graph $W \in \mathbb{R}^{G \times G}$. At training time we do *not* consume all $G$ genes at once. Instead, for each item we sample an index set $I \subseteq \{1, \ldots, G\}$ of size $|I| = s \ll G$ (policy in Sec. A.2 or Algorithm 2). We then restrict expression vectors to this set, $c_x^{(I)} = (c_x(g_i))_{i \in I}$ and $x_t^{(I)} = (x_t(g_i))_{i \in I}$, and extract the masked gene subgraph $M_I = W[I, I]$.

**Gene/context encoding.** The gene encoder $E_g$ consumes the ordered identity sequence $G_I = (g_i)_{i \in I}$ together with mask $M_I$ and produces contextualized gene tokens $Z_I = E_g(G_I; M_I) \in \mathbb{R}^{|I| \times d}$. In parallel, the value embedder $E_v$ maps each scalar expression to $\mathbb{R}^d$ element-wise. We form aligned token sequences

$$h_c = E_v(c_x^{(I)}) + Z_I, \qquad h_v^0 = E_v(x_t^{(I)}) + Z_I,$$

so that identity and value are summed for the *same* genes (index-wise alignment).

**PAD-Transformer block.** Given perturbation embedding $e_p = \mathrm{Emb}(c_p)$ and time embedding $t_{\mathrm{emb}} = \mathrm{MLP}(\mathrm{SinCos}(t))$, PAD-Transformer applies (i) perturbation injection via an adapter, (ii) self-differential attention on the perturbed latent, and (iii) cross-differential attention against $h_c$, with adaLN-Zero modulation by $t_{\mathrm{emb}}$ at every layer. After $L$ layers we decode the velocity $v_\theta$ and form the one-step endpoint approximation

$$\hat{x}_1^{(I)} = x_t^{(I)} + (1 - t)\, v_\theta.$$

**Timestep Embedding.** The timestep $t \in [0, 1]$ is encoded as $t_{\mathrm{emb}} = \mathrm{MLP}(\mathrm{SinCos}(t))$, where $\mathrm{SinCos}(t)$ denotes a sinusoidal embedding of $t$ at multiple frequencies.

$$\mathrm{SinCos}(t) = \big[\sin(\omega_1 t), \cos(\omega_1 t), \ldots, \sin(\omega_k t), \cos(\omega_k t)\big], \tag{14}$$

with $\{\omega_j\}$ a set of predefined frequencies.

**Objective.** The training loss is $\mathcal{L} = \mathcal{L}_{\mathrm{FM}} + \lambda\, \mathcal{L}_{\mathrm{MMD}}$. Conditional Flow Matching $\mathcal{L}_{\mathrm{FM}}$ supervises the instantaneous velocity field along the path $x_t \sim \pi_t(x_0, x_1 \mid c_x, c_p)$. To align terminal *distributions*, we compute $\mathcal{L}_{\mathrm{MMD}} = \mathrm{MMD}^2(\hat{X}_1, X_1)$ on mini-batches of endpoints restricted to $I$. We use a Gaussian RBF *multi-kernel* mixture with *dynamic* bandwidths: at each step a reference scale is estimated from the median of off-diagonal pairwise squared distances within the batch, and a small set of bandwidths $\{\sigma_\ell\}$ is generated by multiplying this scale with fixed factors (e.g., $\{0.5, 1.0, 2.0, 4.0\}$). The unbiased estimator drops self-similarities. Full details appear in Appendix A.2 and Algorithm 1.

**Training procedure.** Algorithm 2 summarizes batching over items, per-item gene-subset sampling, masked gene/context encoding, PAD-Transformer passes, endpoint construction, dynamic kernel selection, and parameter updates. Subset sampling is re-drawn every step unless otherwise noted; fixing the RNG seed and the ordering of $G_I$ ensures reproducibility.

**Inference.** At test time we *only* predict on a selected subset of genes. We choose $I$ using the same policy as in training (e.g., a fixed target subset, or a deterministic sampler), build $Z_I = E_g(G_I; M_I)$ once, and evolve $x^{(I)}$ from $t{=}0$ to $1$ using PAD-Transformer and an ODE stepper (Euler/Heun). The final output is $\hat{x}_1^{(I)}$ on genes $I$ (Algorithm 3). If full-vocabulary outputs are desired, a post-hoc imputation head can be added, but is not used in our experiments.

**Complexity.** Both masked attention in $E_g$ and differential attention in PAD-Transformer scale as $\mathcal{O}(|I|^2)$ per layer; dynamic MMD adds $\mathcal{O}(B^2)$ pairwise evaluations within the batch. Choosing $s{=}|I|$ balances accuracy and compute.

## A.4 EXPERIMENTAL SETUP

### A.4.1 NORMAN DATASET

The Norman dataset is a foundational benchmark for modeling single-cell responses to combinatorial genetic perturbations (Norman et al., 2019). Originating from the work of Norman et al. (2019), the experiment utilized CRISPR activation (CRISPRa) in the K562 human cell line to systematically upregulate target genes. The resulting Perturb-seq data profiles cellular responses to approximately 100 single-gene and 124 dual-gene activations, making it a rich and challenging dataset for evaluating a model's ability to predict complex, combinatorial effects (Norman et al., 2019).

To ensure consistency with recent benchmarks, our study used the publicly available, scFoundation-reprocessed version of the Norman dataset (Ahlmann-Eltze et al., 2025). The specific data file can

---

**Algorithm 2:** Training scDFM (PAD-Transformer) with gene-subset encoding

---

**Input:** Dataset $\mathcal{D} = \{(c_x, x_1, c_p)\}$ with $G$ genes; gene graph $W$; subset sampler $\mathsf{SampleSubset}(G, s)$;
   FM schedule $\pi_t$; MMD scales $\mathcal{S}$; weight $\lambda$; layers $L$

**Output:** Trained parameters $\theta$

1 **for** mini-batch $\mathcal{B} \subset \mathcal{D}$ **do**
2 $\quad$ Sample $t \sim \mathcal{U}(0, 1)$ (per item or per batch);
$\quad$ // Build batch tensors on the same gene index set $I$ per item
3 $\quad$ **foreach** $(c_x, x_1, c_p) \in \mathcal{B}$ **do**
4 $\quad\quad$ $I \leftarrow \mathsf{SampleSubset}(G, s); \quad G_I \leftarrow (g_i)_{i \in I}; \quad M_I \leftarrow W[I, I];$
5 $\quad\quad$ $x_t \leftarrow$ sample from $\pi_t(x_0, x_1 \mid c_x, c_p);$
6 $\quad\quad$ $Z_I \leftarrow E_g(G_I; M_I);$ $\qquad\qquad\qquad$ // contextual gene identities
7 $\quad\quad$ $h_c \leftarrow E_v(c_x^{(I)}) + Z_I; \quad h_v^0 \leftarrow E_v(x_t^{(I)}) + Z_I;$
8 $\quad\quad$ $e_p \leftarrow \mathrm{Emb}(c_p); \quad t_{\mathrm{emb}} \leftarrow \mathrm{MLP}(\mathrm{SinCos}(t));$
9 $\quad\quad$ $h_v^L \leftarrow \text{PAD-Transformer}(h_v^0, h_c, e_p, t_{\mathrm{emb}}; L);$
10 $\quad\quad$ $v_\theta \leftarrow \mathrm{DecodeVelocity}(h_v^L, e_p);$
11 $\quad\quad$ $\mathcal{L}_{\mathrm{FM}} \mathrel{+}= \mathrm{CFM}(v_\theta; x_t^{(I)}, c_x^{(I)}, c_p, t);$
12 $\quad\quad$ $\hat{x}_1^{(I)} \leftarrow x_t^{(I)} + (1-t)\, v_\theta;$
13 $\quad\quad$ collect $\hat{x}_1^{(I)}$ and $x_1^{(I)}$ into batch sets $\hat{X}_1, X_1;$
14 $\quad$ $\Sigma \leftarrow$ dynamic bandwidths via batch median heuristic on $X_1$ (Appendix A.2);
15 $\quad$ $\mathcal{L}_{\mathrm{MMD}} \leftarrow \mathrm{MMD}^2(\hat{X}_1, X_1; \Sigma);$
16 $\quad$ $\mathcal{L} \leftarrow \mathcal{L}_{\mathrm{FM}} + \lambda\, \mathcal{L}_{\mathrm{MMD}}; \quad$ Update $\theta \leftarrow \theta - \eta \nabla_\theta \mathcal{L};$

---

**Algorithm 3:** Inference with PAD-Transformer (gene-subset input)

---

**Input:** Trained $\theta$; control profile $c_x$; condition $c_p$; subset index $I$ (same policy as training); gene graph
   $W$; steps $K$

**Output:** Predicted perturbed expression $\hat{x}_1^{(I)}$ on genes $I$

1 $G_I \leftarrow (g_i)_{i \in I}; \quad M_I \leftarrow W[I, I]; \quad Z_I \leftarrow E_g(G_I; M_I);$
2 Initialize $x_0^{(I)} \sim q_0$ (e.g., noise; or use a standard source);
3 **for** $k = 0$ **to** $K-1$ **do**
4 $\quad$ $t \leftarrow k/K; \quad t_{\mathrm{emb}} \leftarrow \mathrm{MLP}(\mathrm{SinCos}(t)); \quad e_p \leftarrow \mathrm{Emb}(c_p);$
5 $\quad$ $h_c \leftarrow E_v(c_x^{(I)}) + Z_I; \quad h_v^0 \leftarrow E_v(x_k^{(I)}) + Z_I;$
6 $\quad$ $h_v^L \leftarrow \text{PAD-Transformer}(h_v^0, h_c, e_p, t_{\mathrm{emb}}; L);$
7 $\quad$ $v_\theta \leftarrow \mathrm{DecodeVelocity}(h_v^L, e_p);$
8 $\quad$ $x_{k+1}^{(I)} \leftarrow x_k^{(I)} + \Delta t\, v_\theta;$ $\qquad\qquad$ // Euler; Heun/ODE solver optional
9 **return** $\hat{x}_1^{(I)} \leftarrow x_K^{(I)};$ $\qquad$ // prediction is on the selected genes only

---

be downloaded directly from this Figshare link. We processed the data using the following standard pipeline in `Scanpy`:

- **Normalization:** Library size was normalized to a target sum of 10,000 counts per cell using `sc.pp.normalize_total`.

- **Log Transformation:** Expression values were log-transformed with `sc.pp.log1p` to stabilize variance.

- **Gene Selection:** We selected the top 5,000 highly variable genes using `sc.pp.highly_variable_genes`.

This procedure, combined with the inclusion of the perturbed target genes, resulted in a final feature set of 5,029 genes for model training. For evaluation, we focused on the top 1,000 most highly expressed genes, a common practice that ensures the assessment is performed on robust biological signals Ahlmann-Eltze et al. (2025).

To guarantee the robustness of our findings, all experiments were conducted using four independent random train/validation/test splits, and we report the mean performance across these runs. The exact

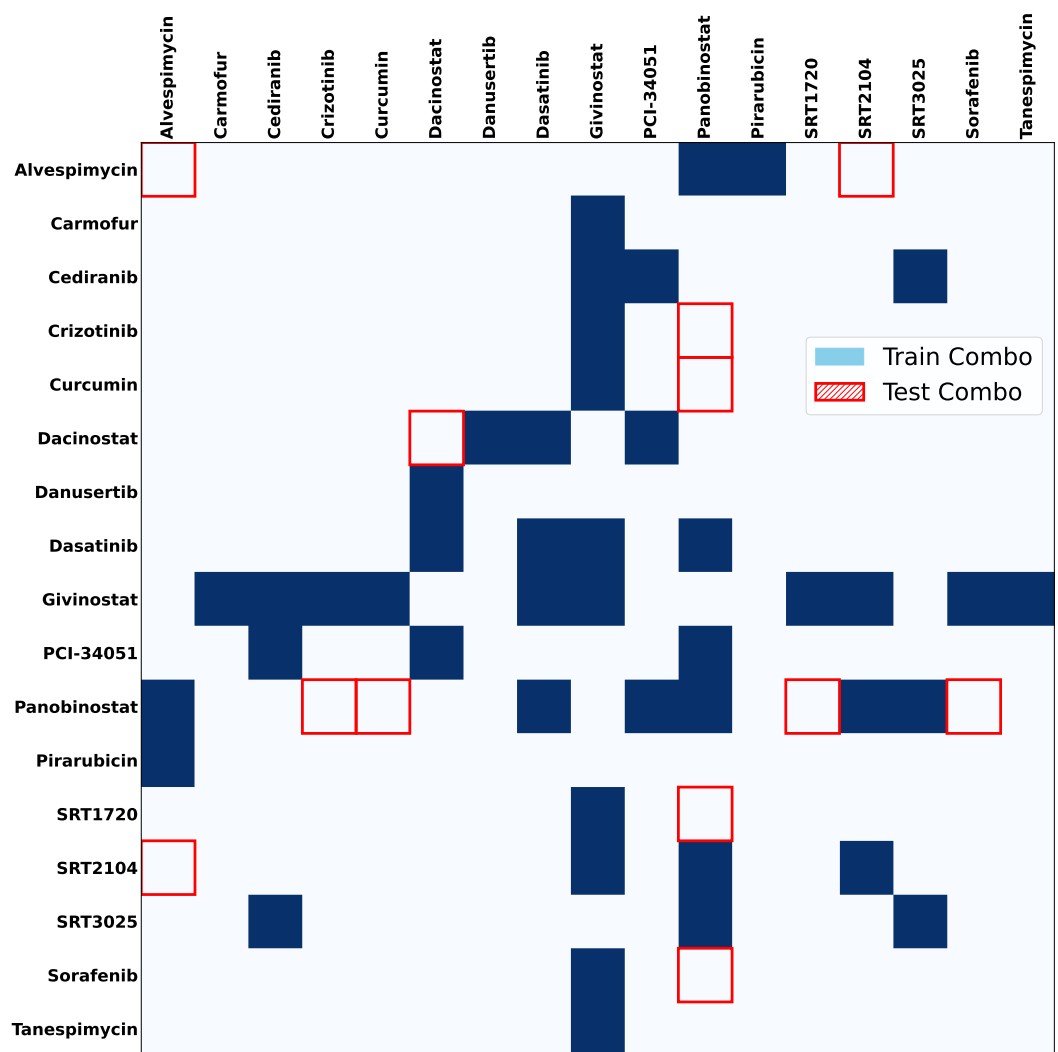

Figure 6: ComboSciPlex combination data split. Rows and columns denote small-molecule compounds. Filled deep blue cells indicate drug pairs used for **training**; red outlined cells denote **held-out test** pairs.

preprocessing configurations and data split indices will be made fully available in our public code repository to ensure complete reproducibility.

### A.4.2 COMBOSCIPLEX

**ComboSciPlex dataset.** We use the ComboSciPlex drug-combination dataset (Lotfollahi et al., 2023), a follow-up extension of the SciPlex chemical barcoding platform (Srivatsan et al., 2020), designed to measure single-cell transcriptional responses to pairwise drug perturbations. In the preprocessed release used by prior work (Klein et al., 2025; Lotfollahi et al., 2023), the dataset contains 63,378 single-cell transcriptomes and a total of 32 treatment conditions (single agents and pairwise combinations) in A549 (see Fig. 6). Programmatic access is available via `pertpy` (`pertpy.data.comboscisplex()`); we additionally mirror the exact files used in our runs (Figshare DOI below). For our experiments, we follow common practice and restrict the expression space to the top 5,000 highly expressed genes after QC/normalization, which has been shown to work well on this dataset family. The train/test split is illustrated in Fig. 6: all available single-agent profiles are included in training, and evaluation is conducted on held-out drug combinations (red outlines) to probe combinatorial generalization; the precise list of train/test pairs and their counts

are released with our code. Downloads: pertpy loader (documentation) and Figshare (preprocessed subset).

### A.4.3 TRAINING SETTING FOR `scDFM`

All experiments were performed on a cluster equipped with NVIDIA H800 GPUs.

**Norman training setup.** We train `scDFM` on the Norman dataset with the following configuration:

- **Optimizer & LR schedule:** Adam with an initial learning rate $5 \times 10^{-5}$, decayed by a cosine schedule to $\eta_{\min} = 10^{-6}$.

- **Batch size:** 96.

- **Training length:** 100,000 optimization steps.

- **Distribution regularization:** MMD loss with weight $\lambda = 0.5$ (dynamic multi-kernel RBF; details in Appendix).

- **Gene–gene mask:** kNN graph with $k = 30$ built from signed (both positive and negative) gene–gene correlations; the resulting mask is used in the gene encoder's self-attention.

- **Perturbation embedding:** for CRISPR activations, the perturbation embedding for gene $g$ shares parameters with the gene identity embedding $E_g(g)$ (no separate perturbation-embedding table).

- **Backbone width/depth:** hidden size $d = 512$, $L = 4$ layers, $H = 8$ attention heads, dropout 0.1 (applied to attention and MLP).

- **Inference:** Euler ODE rollout with $K = 100$ uniform steps over $t \in [0, 1]$ (i.e., $\Delta t = 0.01$).

Unless otherwise noted, results are averaged over four independent random train/val/test splits using the same preprocessing and masking pipeline.

**ComboSciPlex setting (difference from Norman).** Unlike Norman, where the perturbation embedding for gene $g$ shares parameters with the gene identity embedding $E_g(g)$, in ComboSciPlex we use a dedicated perturbation embedding table $E_p$ for small molecules. The drug embeddings in $E_p$ are learned end-to-end and are not tied to gene tokens; multi-drug conditions are embedded via $E_p$ and injected at every layer through the same adapter mechanism as in Norman. All other hyperparameters (optimizer, LR schedule, backbone width/depth, kNN=30 signed gene–gene mask, $\lambda$=0.5 for MMD) follow the Norman setup.

### A.4.4 METRIC DEFINITIONS

To comprehensively assess both pointwise prediction accuracy and distributional fidelity, we report multiple evaluation metrics adapted from prior work in single-cell perturbation modeling. Most implementations follow the standard routines provided in the `celleval` library.[1]

**L2 (Mean-level Perturbation Distance).** To quantify expression deviation at the perturbation level, we compute the average L2 distance between predicted and ground-truth \*\*mean gene expression vectors\*\* for each perturbation. Formally, for each non-control perturbation $p \in \mathcal{P}$, we define:

$$\text{L2}_{\text{mean}} = \frac{1}{|\mathcal{P}|} \sum_{p \in \mathcal{P}} \|\hat{\mu}_p - \mu_p\|_2, \tag{15}$$

where $\mu_p = \frac{1}{N_p} \sum_{i=1}^{N_p} x_i^{(p)}$ and $\hat{\mu}_p = \frac{1}{N_p} \sum_{i=1}^{N_p} \hat{x}_i^{(p)}$ are the empirical mean vectors over $N_p$ cells under perturbation $p$, from ground-truth and predicted expression respectively. This metric evaluates global shifts between predicted and real perturbation responses in gene expression space.

---

[1] https://github.com/ArcInstitute/cell-eval

**MSE, MAE.** These metrics quantify absolute cell-level error between predicted and ground-truth gene expression. Let $\hat{X}, X \in \mathbb{R}^{N \times G}$ be the predicted and real expression matrices. Then:

- **MSE** $= \frac{1}{NG} \sum_{i=1}^{N} \sum_{j=1}^{G} \left( X_{ij} - \hat{X}_{ij} \right)^2$

- **MAE** $= \frac{1}{NG} \sum_{i=1}^{N} \sum_{j=1}^{G} \left| X_{ij} - \hat{X}_{ij} \right|$

**Perturbation Discrimination Score (PDS).** This metric is a rank-based retrieval score based on pseudobulk similarity; it evaluates whether predicted perturbation effects resemble their true counterparts more than other unrelated perturbations.

Let $\hat{X}^k \in \mathbb{R}^{n_k \times G}$ and $X^k \in \mathbb{R}^{m_k \times G}$ denote the predicted and true log-normalized expression matrices under perturbation $k$ (excluding control), with $n_k$ and $m_k$ cells respectively. We first compute pseudobulk vectors by averaging across cells:

$$\hat{y}_k = \frac{1}{n_k} \sum_{i=1}^{n_k} \hat{X}_i^k, \quad y_k = \frac{1}{m_k} \sum_{j=1}^{m_k} X_j^k. \tag{16}$$

For each perturbation $p$, we calculate the $L_1$ distance between its predicted pseudobulk $\hat{y}_p$ and all ground truth pseudobulks $\{y_t\}_{t=1}^{N}$, where $N$ denotes the total number of perturbation categories:

$$d_{pt} = \sum_{g \notin \mathcal{G}_p} |\hat{y}_{p,g} - y_{t,g}|, \tag{17}$$

where $\mathcal{G}_p$ denotes the set of genes directly perturbed by $p$ (which are excluded from the comparison).

We sort $\{d_{pt}\}_{t=1}^{N}$ in ascending order, and record the rank of the true target $t = p$:

$$\text{rank}_p = \arg \text{sort}_t \ d_{pt}. \tag{18}$$

Finally, the discrimination score for perturbation $p$ is defined as:

$$\text{PDS}_p = 1 - \frac{\text{rank}_p - 1}{N - 1}. \tag{19}$$

A perfect match yields $\text{PDS}_p = 1$. The overall score is averaged across all predicted perturbations:

$$\text{PDS} = \frac{1}{N} \sum_{p=1}^{N} \text{PDS}_p. \tag{20}$$

**DE-Spearman-Sig.** To measure biological relevance, we compute the Spearman correlation between predicted and true log fold-changes for genes that are significantly differentially expressed in ground truth (adjusted $p$-value $< 0.05$). This focuses evaluation on meaningful changes and filters out low-signal noise.

**Pearson $\Delta$.** This measures the difference in Pearson correlation matrices between predicted and true cell-wise gene expression. For each gene, the Pearson correlation vector across cells is computed, and the average L1 difference between these vectors defines $\Delta$.

**Pearson $\hat{\Delta}$.** This metric evaluates whether the model captures sample-specific heterogeneity beyond the average perturbation effect, following the principles of the Systema framework. Unlike standard evaluations that use the control group as a baseline, we employ the mean expression of the specific perturbation from the training set as the reference to isolate non-systematic variations. Let $\hat{\mathbf{x}}_i \in \mathbb{R}^G$ and $\mathbf{x}_i \in \mathbb{R}^G$ denote the predicted and true log-normalized expression vectors for cell $i$ under perturbation $p$, respectively. Let $\bar{\mathbf{x}}_{\text{train}}^{(p)} \in \mathbb{R}^G$ represent the centroid (mean expression profile) of perturbation $p$ computed from the training data. We define the residual vectors as:

$$\boldsymbol{\delta}_{i,\text{pred}} = \hat{\mathbf{x}}_i - \bar{\mathbf{x}}_{\text{train}}^{(p)}, \quad \boldsymbol{\delta}_{i,\text{true}} = \mathbf{x}_i - \bar{\mathbf{x}}_{\text{train}}^{(p)}. \tag{21}$$

The Pearson $\hat{\Delta}$ score is computed as the average Pearson correlation coefficient ($\rho$) between these residual vectors across all cells in the test set ($N$):

$$\text{Pearson } \hat{\Delta} = \frac{1}{N} \sum_{i=1}^{N} \rho(\boldsymbol{\delta}_{i,\text{pred}}, \boldsymbol{\delta}_{i,\text{true}}). \tag{22}$$

A higher score indicates that the model successfully predicts the specific cellular response variations that deviate from the population mean.

**Pearson $\hat{\Delta}_{20}$.** To focus the evaluation on the most biologically relevant variations and mitigate the impact of noise from invariant genes, this metric restricts the Pearson $\hat{\Delta}$ calculation to the top-20 genes with the highest variance. Let $\mathcal{G}_{20}$ denote the subset of 20 genes exhibiting the highest variance in the true residuals ($\boldsymbol{\delta}_{\text{true}}$) across the perturbation population. The metric is defined as:

$$\text{Pearson } \hat{\Delta}_{20} = \frac{1}{N} \sum_{i=1}^{N} \rho\left(\boldsymbol{\delta}_{i,\text{pred}}[\mathcal{G}_{20}], \boldsymbol{\delta}_{i,\text{true}}[\mathcal{G}_{20}]\right), \tag{23}$$

where $[\mathcal{G}_{20}]$ denotes indexing the vectors to include only the genes in the subset. This serves as a stricter test of the model's precision in capturing high-variance gene programs.

## A.5 BASELINES

**Control (no-change).** A naïve identity baseline that predicts no perturbation effect:

$$\hat{x} = c_x,$$

i.e., the post-perturbation profile is taken to be the pre-perturbation control profile.

**Additive (linear superposition).** For a combination of two single perturbations $a$ and $b$, we approximate the combined effect as the sum of single-agent deltas measured (or predicted) relative to control:

$$\hat{x}_{a+b} = c_x + \left(x^{(a)} - c_x\right) + \left(x^{(b)} - c_x\right),$$

where $x^{(a)}$ and $x^{(b)}$ are the single-perturbed profiles (when available, from the same split) aligned to the same gene index as $c_x$. This baseline encodes a strictly additive interaction model without higher-order or non-linear effects.

**scGPT (Transformer without pretraining in our setting).** A gene-token Transformer that represents a cell as a sequence of gene tokens with value and identity embeddings and predicts the perturbed expression conditioned on $c_x$ and $c_p$. In our experiments, *we train scGPT from scratch on our splits* (no external pretraining), keeping architecture capacity comparable to ours and injecting $c_p$ at the input/adapter layers for conditioning.

**Geneformer (pretrained foundation model).** A large pretrained Transformer for single-cell transcriptomics (pretrained on large corpora of scRNA-seq profiles). We fine-tune the released checkpoint on our task/splits to map $(c_x, c_p)$ to post-perturbation expression. Concretely, perturbation information is injected via InSilicoPerturber, and then we map the CLS token's representation to perturbed expression with linear probe.

**GEARS (graph-based perturbation predictor).** A graph-aware baseline that encodes intergene structure (e.g., coexpression/regulatory graphs) via message passing and predicts gene-level responses under single and combinatorial perturbations. Combinations are modeled by jointly conditioning on multiple targets in the graph encoder/decoder. We use the official implementation and default training recipe adapted to our preprocessing and splits.

**CPA (Compositional Perturbation Autoencoder).** A compositional latent-space model in which an encoder maps $c_x$ to a latent representation, learned embeddings encode perturbation (and optional covariates/dose), and a decoder reconstructs the perturbed state. Combination treatments are composed by adding condition embeddings in latent space (plus optional dose scalings), enabling zero-shot composition. We follow the public implementation with our data normalization and split protocol.

**State (State Transition Model).** We followed the official State implementation and trained the model from scratch using the publicly released code (Adduri et al., 2025). We did not use the pretrained State embedding model or the pretrained State transition model for two reasons. First, the released pretrained State transition weights are designed specifically for the "drug type + dosage" formulation and do not support drug combinations or gene perturbations, which is the evaluation setting of our work. Second, while pretrained embeddings are provided, no fine-tuning pipeline is included in the original release, and adapting the pretrained embedding model to the combination-prediction setting would require reimplementation of components that were not publicly available. Training from scratch therefore ensured a fair and reproducible experimental setup aligned with the setting of our benchmarking experiments.

**CellFlow.** CellFlow (Klein et al., 2025) models perturbation response as a continuous transformation between control and perturbed cell states in a low-dimensional latent space, where a flow-matching objective is used to learn the velocity field conditioned on perturbation identity. During inference, the model integrates the learned flow starting from a control cell to obtain its predicted perturbed state. In our implementation, following the official open-source reference, we first reduce gene expression to a 50-dimensional PCA space, which serves as the latent representation for both control and perturbed profiles. The predicted latent state is subsequently mapped back to full gene space using the PCA decoder for evaluation. All hyperparameters and data preprocessing steps follow the public implementation unless otherwise specified to ensure strict comparability with reported results.

**Implementation note.** All baselines are trained/evaluated under the same pre-processing, gene index alignment, and four random train/val/test splits as our method (means reported).

### A.6 DISCUSSION ABOUT LIMITATIONS AND FUTURE WORKS

**Distributional Flow Matching.** Our results suggest that enforcing distribution-level alignment is a core requirement for modeling realistic perturbation effects. Beyond biology, the idea of distribution-aware flow matching may also benefit tasks in vision and generative modeling. Exploring this direction remains an exciting topic for future work.

**Representation and Path Design.** In this work, we adopt a simple linear interpolant in the log-normalized gene expression space as the default reference path for flow matching. While this choice yields stable training and interpretable trajectories, it is arguably suboptimal: biological processes may follow nonlinear, branching, or manifold-constrained transitions that are poorly approximated by linear paths in log space. We believe future research should explore both the choice of representation space and the design of interpolation trajectories. A more principled understanding of which interpolants best capture perturbation dynamics could significantly enhance the realism and generalization of generative perturbation models. However, we leave these directions for future exploration.

**Scalability to Multi-Context Datasets.** Our study focuses on two representative datasets, Norman and ComboSciPlex, but does not include more recent large-scale resources such as ARC-state (Adduri et al., 2025), which became available during our work. These datasets span multiple cell lines and perturbation types, offering a more comprehensive benchmark for generalization. Extending our method to such diverse contexts will be important for future work.

**Structural Priors and Graph Topology.** We utilized a Pearson correlation-based graph to construct the attention mask, serving as a computational proxy for gene regulatory networks. While this approach effectively introduces biological structure, it primarily captures linear co-expression patterns and may miss complex non-linear or causal dependencies. We view `scDFM` as a flexible framework where the graph topology is a modular component. Future iterations could significantly benefit from incorporating more sophisticated priors, such as those derived from Mutual Information (MI) or causal discovery algorithms, to better capture the non-linear dynamics of gene regulation.

**Scalability to Multi-Context Datasets.** Our study currently focuses on representative benchmarks for genetic and drug perturbations. A critical future direction is applying scDFM to broader, large-

Table 4: **Additive setting** — variance across four random folds.

| Model | L2 | MSE | MAE | DE-Spearman $\rho$ | Pearson $\Delta$ | DS |
|---|---|---|---|---|---|---|
| Control | 0.13003 | 0.00100 | 0.00100 | N.A. | N.A. | 0.00297 |
| Additive | 0.03077 | 0.00015 | 0.00015 | 0.04000 | 0.01090 | 0.00899 |
| scGPT | 0.10928 | 0.00096 | 0.00096 | 0.20351 | 0.02374 | 0.00811 |
| Geneformer | 0.07275 | 0.00045 | 0.00133 | 0.13790 | 0.02357 | 0.03896 |
| GEARS | 0.16239 | 0.00133 | 0.00133 | 0.07530 | 0.02577 | 0.01582 |
| CPA | 0.26981 | 0.00301 | 0.00301 | 0.08628 | 0.00357 | 0.04691 |
| Ours | 0.04685 | 0.00023 | 0.00023 | 0.12040 | 0.01278 | 0.01153 |

scale initiatives such as the Virtual Cell Challenge (Roohani et al., 2025). Validating the model on such massive resources, spanning multiple cell lines and diverse perturbation mechanisms like CRISPRi/a, will be crucial for demonstrating robustness across heterogeneous biological contexts and advancing foundation-scale perturbation modeling.

## A.7 CASE STUDY

**Case Study: *CEBPE+CEBPA*.** To better illustrate model behavior, we analyze the joint perturbation of *CEBPE* and *CEBPA*. For this case study, we first selected the top 20 genes showing the largest absolute expression changes relative to the control state. For each gene, we plotted the distribution of expression changes across single cells as boxplots (after subtracting the control mean), and overlaid the model predictions as blue dots representing the mean predicted shift. This setup enables a direct comparison between empirical distributions and different predictive models.

As shown in Fig. 7 and Fig. 8, the seven panels cover all baselines considered: Additive, Control, CPA, GEARS, Geneformer, scGPT, and `scDFM`. The control and additive baselines capture only coarse global trends and fail to reproduce non-linear responses. CPA and GEARS improve alignment but still deviate on synergistic targets. Geneformer and scGPT provide partial improvements, yet often misestimate magnitudes or underperform on non-additive regulation. By contrast, our framework consistently places predictions within the observed variance and faithfully recovers synergistic upregulation in genes such as *CEACAM20* and *LST1*. Together, these results confirm the necessity of explicitly modeling gene–gene interactions and highlight the advantage of our approach in capturing complex combinatorial perturbation effects.

## A.8 CONSISTENCY UNDER RANDOMIZED EVALUATION

**Variance across random folds.** Tables 4 and 5 report the variance of each metric across four random splits for the additive and holdout (single/double) settings, respectively (values rounded to five decimals). Lower variance indicates greater run-to-run stability. In the additive setting, methods that rely on simple composition (e.g., the Additive baseline) unsurprisingly exhibit small variance, while learned baselines such as CPA and GEARS show substantially larger variability. Our `scDFM` maintains consistently low and competitive variance across metrics (e.g., small L2/MSE/MAE and modest discrimination score), indicating stable training despite modeling the full conditional distribution. In the holdout regime, variances increase for all methods—more so for the double setting—reflecting the higher difficulty of extrapolating to fully unseen combinations. `scDFM` remains competitive (e.g., second–lowest L2 variance under the double setting) and markedly more stable than CPA, which exhibits the largest variability. Entries marked "N.A." correspond to metrics that are undefined for a given baseline (e.g., Pearson $\Delta$ for Control). Overall, these results complement the main-text means by showing that `scDFM` delivers not only strong accuracy but also robust behavior under random data splits.

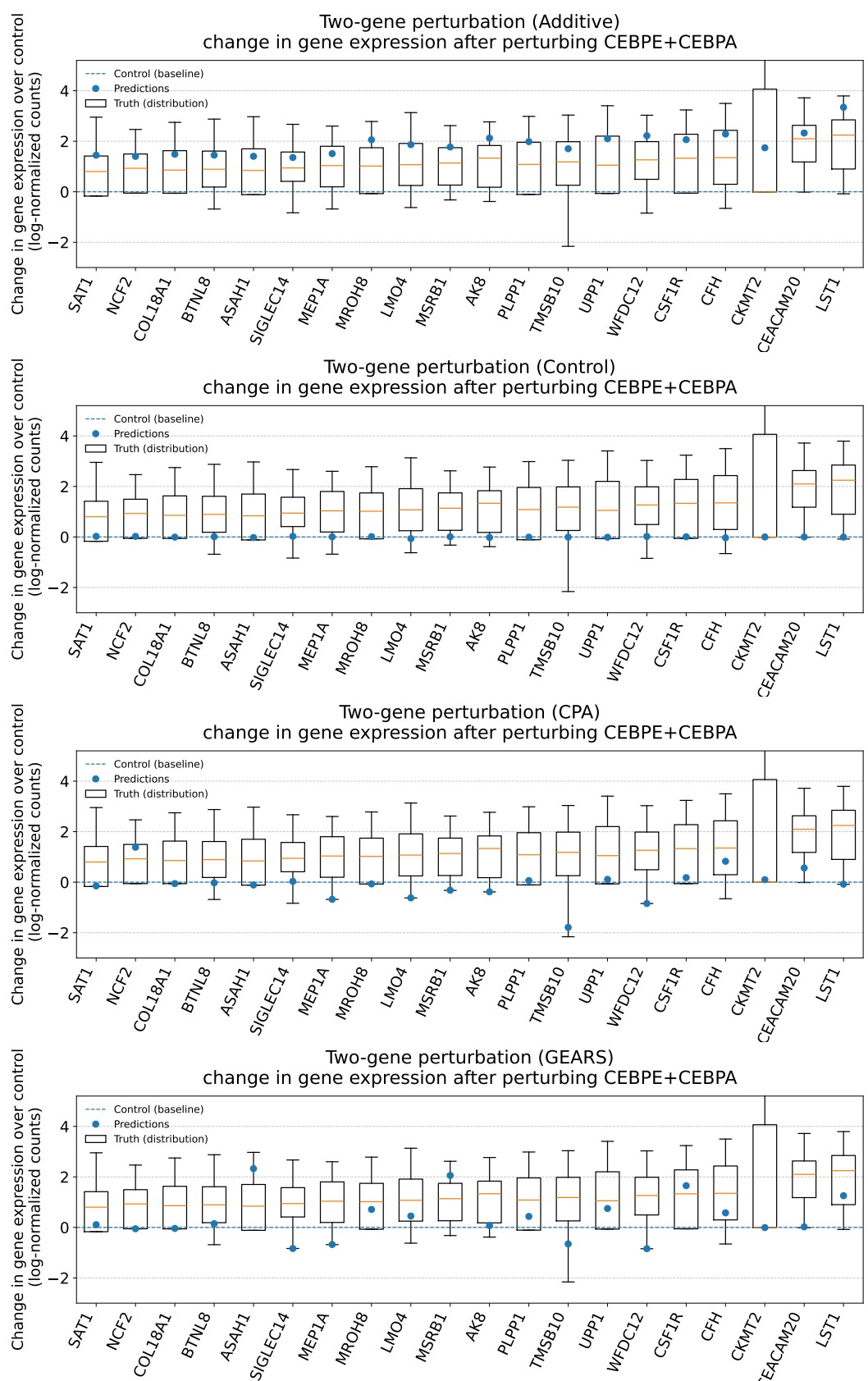

Figure 7: Comparison across additive, control, CPA and GEARS cases.

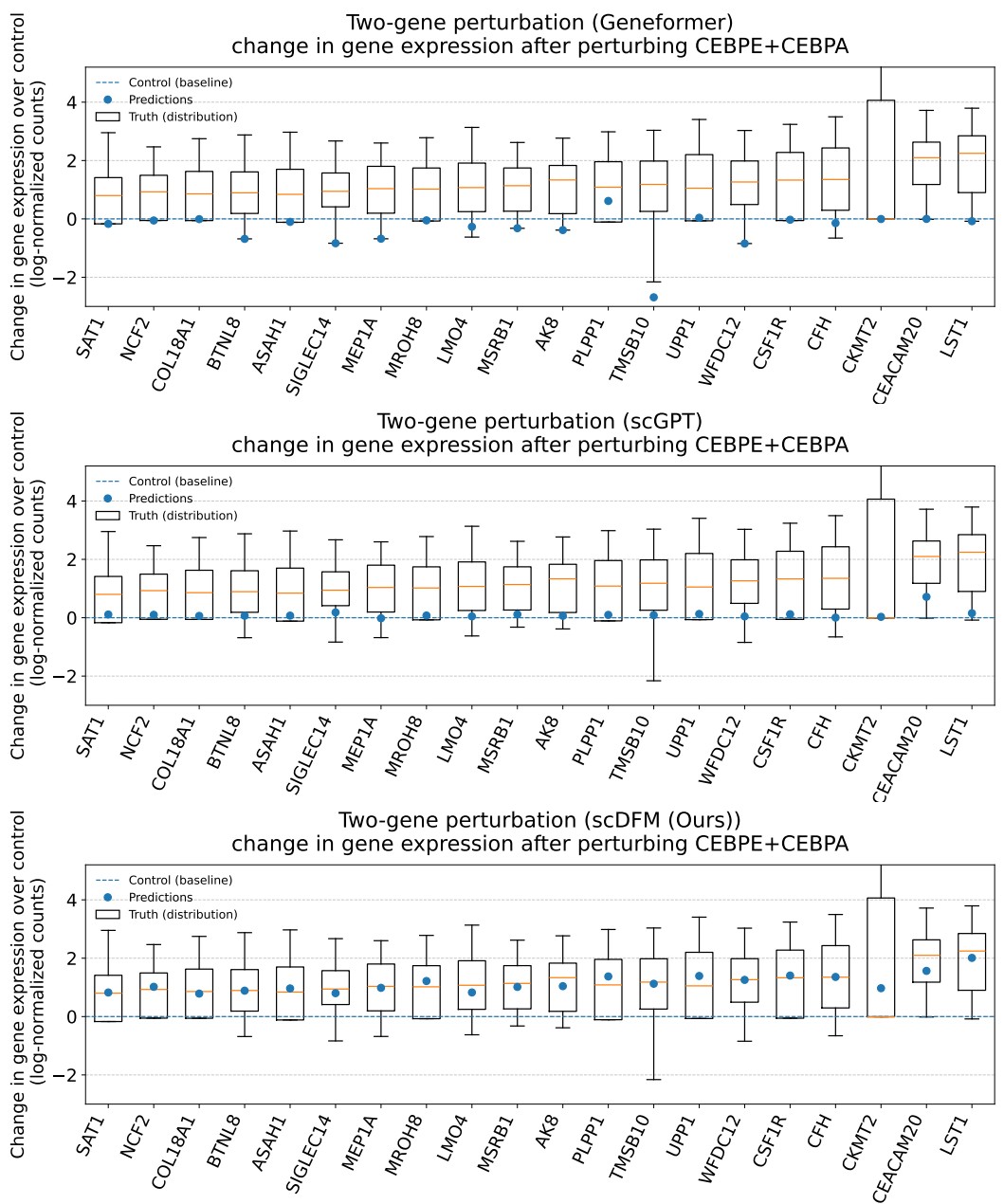

Figure 8: Comparison of Geneformer, scGPT, and **scDFM** (Ours) on the CEBPE+CEBPA case.

Table 5: **Holdout setting** — variance across four random folds.

| Setting | Model | L2 | MSE | MAE | DE-Spearman $\rho$ | Pearson $\Delta$ | DS |
|---|---|---|---|---|---|---|---|
| Single | Control | 0.17045 | 0.00040 | 0.00103 | N.A. | N.A. | 0.00244 |
| | scGPT | 0.08057 | 0.00035 | 0.00115 | 0.09429 | 0.03844 | 0.01939 |
| | Geneformer | 0.15349 | 0.00056 | 0.00128 | 0.13532 | 0.05961 | 0.06346 |
| | GEARS | 0.14342 | 0.00067 | 0.00213 | 0.22257 | 0.01262 | 0.02299 |
| | CPA | 0.66121 | 0.00678 | 0.01183 | 0.28781 | 0.02085 | 0.06246 |
| | **scDFM (ours)** | 0.10947 | 0.00042 | 0.00036 | 0.16871 | 0.03498 | 0.03304 |
| Double | Control | 0.26210 | 0.00242 | 0.00323 | N.A. | N.A. | 0.00192 |
| | scGPT | 0.22748 | 0.00208 | 0.00330 | 0.23503 | 0.05280 | 0.00667 |
| | Geneformer | 0.33132 | 0.00160 | 0.00320 | 0.28817 | 0.05467 | 0.07668 |
| | GEARS | 0.26108 | 0.00215 | 0.00474 | 0.23465 | 0.01816 | 0.01295 |
| | CPA | 0.93954 | 0.01017 | 0.01746 | 0.14223 | 0.03166 | 0.04369 |
| | **scDFM (ours)** | 0.24457 | 0.00128 | 0.00218 | 0.10788 | 0.01025 | 0.01895 |

