# OpenReview forum: "scDFM: Distributional Flow Matching Model for Robust Single-Cell Perturbation Prediction"
_ICLR.cc/2026/Conference — ICLR 2026 Poster_

### Official Review · Reviewer_JmMZ · 2025-10-25

**Soundness:** 3
**Presentation:** 3
**Contribution:** 2
**Rating:** 4
**Confidence:** 4

**Summary:**

scDFM is a generative framework for single-cell perturbation prediction based on conditional flow matching. To model population-level effects, it combines a flow matching objective with a Maximum Mean Discrepancy (MMD) loss. The model's backbone, the Perturbation-Aware Differential Transformer (PAD-Transformer), incorporates a gene co-expression graph to guide its attention mechanism. The method is evaluated on combinatorial genetic and drug perturbation datasets.

**Strengths:**

- The combination of flow matching for trajectory modeling and MMD loss for distributional alignment is a sound approach for this task.

- The PAD-Transformer architecture, which incorporates biological priors via a co-expression graph, is an interesting design choice.

- The evaluation incorporates several cell-eval metrics which is encouraged.

**Weaknesses:**

- MMD has already been considered in STATE [1], and flow-matching has already been covered in CellFlow [2]. Therefore the novelty needs to be carefully positioned. It is unclear why performing flow matching in the original space is a meaningful novelty and improvement over e.g. cell-flow.
- It is unclear how the results compare to CellFlow, given its similarity. Also, the paper’s citation on CellFlow [2] seems wrong.
- It is unclear how multi-step generation increases performance.
- There seems to be a discrepancy from [3]. In particular, in [3], the additive baseline performs better than Geneformer, but Fig. 3 here seems to show an opposite trend. Furthermore, it is unclear how the results compare to well-established benchmarks in [3] on unseen perturbation and drug perturbations.
- The intuition on the gene attention mask remains unclear. The exact same argument could be applied to image or language data, but bi-directional attention without masking is still widely adopted.

[1] Adduri, Abhinav K., et al. "Predicting cellular responses to perturbation across diverse contexts with State." bioRxiv(2025): 2025-06.
[2] Klein, Dominik, et al. "CellFlow enables generative single-cell phenotype modeling with flow matching." bioRxiv (2025): 2025-04.
[3] Ahlmann-Eltze, Constantin, Wolfgang Huber, and Simon Anders. "Deep-learning-based gene perturbation effect prediction does not yet outperform simple linear baselines." Nature Methods (2025): 1-5.

**Questions:**

See weaknesses.

---

> ### Author Response · Authors · 2025-11-23
> **Official Response to Reviewer JmMZ (1)**
>
> We thank the reviewer for the helpful feedback and thoughtful questions. The comments on novelty positioning, comparisons, and design choices are appreciated, and we have addressed each point below. Corresponding revisions are marked in blue in the updated manuscript.
>
> ---
>
> > **W1**. MMD has already been considered in STATE [1], and flow matching has already been covered in CellFlow [2]. Therefore the novelty needs to be carefully positioned. It is unclear why performing flow matching in the original space is a meaningful novelty and improvement over e.g. cell-flow.
>
> We thank the reviewer for raising this point. While our method builds upon flow matching and MMD, the novelty does not lie in reusing these components, but in how they are reformulated for the unpaired perturbation setting and integrated into a biologically grounded modeling framework.
>
> - Why flow matching in the original gene space (vs. CellFlow latent space):
>
> CellFlow applies Flow Matching in a low-dimensional latent space obtained via PCA, requiring a projection–reconstruction pipeline. This introduces an information bottleneck where subtle yet biologically meaningful perturbation signals are lost during encoding. Operating directly in the original gene space avoids reconstruction artifacts and preserves gene identity, enabling us to incorporate biologically motivated priors (e.g., gene-gene interaction masking), which are not definable once the representation is compressed. Empirically, this design yields consistently better perturbation recovery and discrimination scores than CellFlow (see **reply to W2 below**).
>
> - Why our use of MMD is fundamentally different from STATE:
>
> STATE applies MMD as its sole training objective in a one-step regression formulation, meaning the model **directly** maps control cells to perturbed cells without modeling a trajectory. In contrast, our framework combines continuous flow matching with an MMD boundary constraint, which provides additional structure necessary for learning **transport** from unpaired distributions. This ensures the trajectory converges to the correct perturbed manifold rather than collapsing toward averaged solutions. As shown in **Tables 1 and 2 of the revised manuscript**, this design yields substantial empirical improvements over STATE across all major metrics, especially in robustness and discrimination under held-out combinatorial perturbation.
>
> Table 1: Performance comparison between STATE and scDFM on the Norman additive setting.
>
> | Model | L2 ↓ | MSE ↓ | MAE ↓ | DE-Spearman ρ ↑ | Pearson Δ ↑ | DS ↑ | Pearson $\hat{\Delta}$ ↑ | Pearson $\hat{\Delta}_{20}$ ↑ |
> |-------|------|--------|--------|-------------|---|--------|------------------|---------|
> | STATE | 17.3330 | 0.30059 | 0.24705 | 0.5288 | -0.0108 | 0.5135 | -0.0069 | 0.2515 |
> | scDFM (Ours) | **1.7043** | **0.00315** | **0.02155** | **0.5705** | **0.8853** | **0.9737** | **0.8468** | **0.9260** |
>
>
> We hope this clarifies that our contribution is not an incremental reuse of prior components, but a reformulation of flow matching for unpaired single-cell perturbation modeling with biologically meaningful structural constraints. By operating directly in gene space rather than a latent embedding, the model preserves gene identity and allows downstream discovery of perturbation-specific regulatory patterns that would be inaccessible in latent-space formulations.
>
> ---
>
> > **W2**. It is unclear how the results compare to CellFlow, given its similarity. Also, the paper’s citation on CellFlow [2] seems wrong.
>
> Thank you for pointing this out. We have corrected the citation and now include CellFlow as a baseline in our experiments (**line 106 in the revised manuscript**). While CellFlow and scDFM both use flow matching, their modeling assumptions differ in two important ways. CellFlow operates in a compressed latent representation (PCA/AE), whereas scDFM performs flow matching directly in gene space and incorporates an MMD-based terminal constraint to handle unpaired marginal distributions.
>
> To help with comparison, here is a short summary of the results on Norman Additive data (full results are provided in **Tables 1 and 2 of the revised manuscript**):
>
> | Model | L2 ↓ | Pearson Δ ↑ | DS ↑ | Pearson $\hat{\Delta}$ ↑ | Pearson $\hat{\Delta}_{20}$ ↑ |
> |--------|------|----|------|----------------|--------|
> | scGPT (single-step) | 2.5007 | 0.4503 | 0.5680 | 0.0747 | 0.0798 |
> | CellFlow (latent FM) | 1.6758 | 0.7109 | 0.8072 | 0.6138 | 0.6753 |
> | Ours (gene-space FM) | **1.6186** | **0.7127** | **0.8914** | **0.6659** | **0.8116** |
>
> CellFlow performs strongly and ranks as the second-best method across most metrics, confirming it is a competitive Flow Matching baseline. However, our method further improves both prediction accuracy and robustness, particularly in discrimination and Systema-style evaluation scores, demonstrating the benefit of operating directly in gene space with distribution-anchored training.

---

> ### Author Response · Authors · 2025-11-23
> **Official Response to Reviewer JmMZ (2)**
>
> > **W3**. It is unclear how multi-step generation increases performance.
>
> We thank the reviewer for the question. The reason multi-step generation improves performance is not architectural coincidence, but a modeling principle: predicting the perturbed state in a single step is a harder problem than refining it gradually.
>
> A one-shot model must directly learn the full nonlinear transformation from baseline to post-perturbation expression, which often leads to oversmoothing or averaging effects because the model compresses complex biological variation into a single projection. In contrast, a multi-step formulation decomposes the transition into several smaller refinements, allowing the model to adjust cell states incrementally and preserve heterogeneity rather than collapsing towards a mean.
>
> Consistent with this intuition, models relying on single-step prediction (such as scGPT) perform substantially worse across reconstruction and perturbation metrics. As shown above, our multi-step flow-based model improves both global accuracy (L2: 3.41 → 1.70) and perturbation signal recovery (Pearson Δ: 0.53 → 0.89), reflecting the benefit of iterative refinement over direct prediction (on Norman additive dataset).
>
> | Model | L2 ↓ | Pearson Δ ↑ | Discrimination Score ↑ |
> |--------|------|--------------|-------------------------|
> | scGPT (single-step) | 3.41 | 0.53 | 0.54 |
> | Ours (multi-step flow) | **1.7** | **0.89** | **0.97** |
>
> ---
>
> > **W4**. There seems to be a discrepancy from [3]. In particular, in [3], the additive baseline performs better than Geneformer, but Fig. 3 here seems to show an opposite trend. Furthermore, it is unclear how the results compare to well-established benchmarks in [3] on unseen perturbations and drug perturbations.
>
> We thank the reviewer for pointing out this comparison. We address the performance discrepancy and the comparison against established benchmarks in two parts:
>
> Firstly, the difference arises from **different dataset splits**. Ahlmann-Eltze, et al. [3] uses the scFoundation split: all singles in training, doubles split 50/50 and repeated 5 times. We use a 70/30 split over all doubles, which is an easier regime and naturally reduces the margin between Additive and Geneformer.
>
> Secondly, in our runs, Geneformer exhibits substantially higher standard deviation than the additive baseline. The Additive baseline has very low variance, while Geneformer’s variance is more than double. This explains why, under certain random splits, Geneformer may appear worse than the Additive baseline.
>
> **Table: mean and standard deviation of L2 metrics on different splits**
>
> | Model | L2 (mean) ↓ | L2 (std) ↓ |
> |--------|------------|------------|
> | Control | 3.9937 | 0.1300 |
> | Additive | 1.9395 | **0.0308** |
> | scGPT | 3.4112 | 0.1093 |
> | Geneformer | 1.9132 | 0.0727 |
> | GEARS | 3.5531 | 0.1624 |
> | CPA | 5.7629 | 0.2698 |
> | Ours | **1.7043** | 0.0469 |
>
> We also highlight that scDFM shows the lowest variances among deep learning models, demonstrating that our method is robust and insensitive to randomness in the split.

---

> ### Author Response · Authors · 2025-11-23
> **Official Response to Reviewer JmMZ (3)**
>
> > **W5**. The intuition on the gene attention mask remains unclear. The exact same argument could be applied to image or language data, but bi-directional attention without masking is still widely adopted.
>
> We thank the reviewer for this comparison. While unmasked attention is standard in NLP/CV, applying it directly to genes overlooks a **critical topological difference**: Images and text have inherent spatial/sequential order; gene expression is an unordered set.
> We argue that the gene attention mask is necessary for two specific reasons:
>
> 1. In NLP/CV, model relies on **Positional Encodings (PE)** to understand structure (e.g., word order or pixel coordinates). Without PE, a Transformer sees a "bag of words." However, in scRNA-seq data, genes are permutation-invariant (no natural sequence). Therefore, we cannot use standard PE. Instead, the gene attention mask serves as a "biological positional encoding." It injects necessary structural information (prior interaction logic) so the model doesn't have to learn topology from scratch from noisy data.
>
> 2. **It is crucial to clarify that we only apply this hard mask in encoder initialization**, not throughout the entire generative process. Encoder uses the mask to provide a biologically grounded starting point (fast convergence) and transformer backbone operates with unmasked/global attention.
>
> This hybrid design gives us the best of both worlds: the mask prevents overfitting noise in the early stages, while the unmasked backbone allows the model to "break free" from the prior and discover novel, perturbation-specific interactions.
>
> To further support this design choice, we performed **an ablation experiment below**, comparing our model with and without the biological mask under identical training conditions. **Removing the mask consistently degrades performance**.
>
> | Variant | L2 ↓ | MSE ↓ | MAE ↓ | Pearson Δ ↑ | Discrimination Score ↑ |
> |---------|------|--------|--------|--------------|-------------------------|
> | w/o Mask | 1.7789 | 0.002859 | 0.01928 | 0.7097 | 0.8867 |
> | Ours (with Mask) | **1.6143** | **0.002657** | **0.01896** | **0.7288** | **0.9016** |
>
> ---
>
> [1] Adduri, Abhinav K., et al. "Predicting cellular responses to perturbation across diverse contexts with State." bioRxiv(2025): 2025-06.
>
> [2] Klein, Dominik, et al. "CellFlow enables generative single-cell phenotype modeling with flow matching." bioRxiv (2025): 2025-04.
>
> [3] Ahlmann-Eltze, Constantin, Wolfgang Huber, and Simon Anders. "Deep-learning-based gene perturbation effect prediction does not yet outperform simple linear baselines." Nature Methods (2025): 1-5.

---

> > ### Comment · Reviewer_JmMZ · 2025-11-25
> >
> > Thanks for the rebuttal, the new experimental results look solid. Nevertheless, the result does not show striking improvement over e.g. additive baseline. In W3 I meant using the scDFM model for one-step generation. I also hold my opinion on W1 and W5, as you are not using true gene regulation but some heuristics to define the gene graph, therefore I really believe you can achieve very similar results through e.g. masking on the representation space. Altogether I decided to increase my score to borderline accept.

---

> > > ### Author Response · Authors · 2025-11-25
> > > **Acknowledgment**
> > >
> > > Thank you very much for the update and for reconsidering your evaluation. We truly appreciate the time and effort you invested in reviewing our work. Your feedback significantly improved the clarity and completeness of the paper.

---

### Official Review · Reviewer_u6LS · 2025-10-29

**Soundness:** 2
**Presentation:** 4
**Contribution:** 2
**Rating:** 2
**Confidence:** 4

**Summary:**

This paper proposes scDFM, a conditional flow-matching model for single-cell perturbation prediction. The method extends standard flow-matching by adding a Maximum Mean Discrepancy (MMD) loss to align generated and empirical cell-state distributions and introduces a Perturbation-Aware Differential Transformer (PAD-Transformer) with a gene–gene co-expression attention mask. The model is evaluated on the Norman and ComboSciPlex datasets in both additive and hold-out regimes. Results show modest improvements over previous methods such as GEARS, CPA, Geneformer, and scGPT.

**Strengths:**

1. The paper is well-written and clear, with a strong visual presentation and reproducibility statement.
2. Applying flow matching directly in the expression space is a reasonable and technically clean adaptation of continuous generative models to the single-cell domain.
3. The idea of incorporating a population-level regularizer (MMD) is conceptually sound and aligns with the motivation to capture distributional, rather than per-cell, perturbation effects.
4. The inclusion of differential attention and gene-graph masking demonstrates awareness of biological structure.

**Weaknesses:**

1. Limited novelty and incremental contribution

Flow matching for biological state modeling has already appeared in multiple works. The MMD term is a straightforward sample-based regularizer with no new theoretical or algorithmic insight. The PAD-Transformer largely reuses existing building blocks (Differential Transformer + GEARS-style gene-masking). As a result, the paper reads as a combination of known components rather than a fundamentally new modeling principle.

2. Outdated benchmarking and missing key baselines

The evaluation follows the GEARS-style additive/holdout split, which has since been shown to be insufficient due to expression-vector overlap between training and test sets. Recent frameworks such as PertEval and Systema explicitly address these issues; none of these are considered here. Furthermore, the paper cites but does not evaluate against the most relevant contemporary models (CellFlow, State), making it difficult to assess progress over the true state of the art.

3. Lack of analysis of computational cost and scalability

Flow matching and MMD introduce non-trivial overhead compared to autoencoder or diffusion models, yet there is no runtime or memory comparison. Without such analysis, it is unclear whether the method is practical for larger or multi-tissue datasets.

**Questions:**

See my weaknesses

---

> ### Author Response · Authors · 2025-11-23
> **Official Response to Reviewer u6LS (1)**
>
> We thank the reviewer for the time and thoughtful evaluation of our work. We appreciate the constructive comments regarding novelty, benchmarking, and scalability. These points helped us strengthen the manuscript. Below, we address each concern in detail, with corresponding revisions highlighted in blue in the updated version.
>
> > W1: **Limited novelty and incremental contribution.**
> >
> > Flow matching for biological state modeling has already appeared in multiple works. The MMD term is a straightforward sample-based regularizer with no new theoretical or algorithmic insight. The PAD-Transformer largely reuses existing building blocks (Differential Transformer + GEARS-style gene-masking). As a result, the paper reads as a combination of known components rather than a fundamentally new modeling principle.
>
> We thank the reviewer for this critical assessment. However, we clarify that our novelty lies not in the invention of the individual components (FM, MMD, Transformer), but in tailoring and unifying them to solve a specific, fundamental challenge in single-cell perturbation modeling: the "Unpaired Inference" problem.
>
> Standard Flow Matching assumes access to paired data or relies on simple optimal transport assumptions that may not hold biologically. Single-cell perturbation data, however, provides only independent marginal distributions without cell-to-cell correspondence. Simply applying off-the-shelf FM in this context often fails to capture population-level shifts correctly (see **Fig. 4 in the manuscript**). To address this, we introduce an MMD-based distributional boundary constraint: FM learns the local dynamics, while MMD ensures the trajectory reaches the correct perturbed population at t=1. In our setting, MMD is not a minor regularizer but a necessary mechanism to resolve the underdetermined nature of vector field learning from unpaired data.
>
> In addition, we operate directly in gene space, which allows us to preserve biological structure and incorporate gene-level priors. Ablations show that removing these choices (gene-gene interaction or differential attention) substantially reduces generalization. (see **Fig. 5 in the manuscript**).
>
> Empirically, the framework achieves consistent improvements over CellFlow [1], STATE [4], scGPT [3], and other baselines across evaluated metrics (see **Tables 1 and 2 of the revised manuscript**), supporting the necessity of the design rather than it being a simple combination of known techniques.
>
> **Our key contribution is demonstrating that unpaired flow matching in gene space and anchored by distributional constraints can recover meaningful biological perturbation dynamics and generalize to unseen drug combinations, outperforming both latent flow models and pretrained single-step predictors.**

---

> ### Author Response · Authors · 2025-11-23
> **Official Response to Reviewer u6LS (2)**
>
> > W2: **Outdated benchmarking and missing key baselines.**
> >
> > The evaluation follows the GEARS-style additive/holdout split, which has since been shown to be insufficient due to expression-vector overlap between training and test sets. Recent frameworks such as PertEval and Systema explicitly address these issues; none of these are considered here. Furthermore, the paper cites but does not evaluate against the most relevant contemporary models (CellFlow, State), making it difficult to assess progress over the true state of the art.
>
>
> Thanks for the comment. **First**, our evaluation setting follows the widely used GEARS-style additive and holdout splits, which remain the standard benchmark across perturbation prediction work (including GEARS [5], CPA [6], scGPT [3], Geneformer [2], CellFlow [1], etc.). Using this split ensures comparability with prior literature rather than being a limitation of our evaluation.
>
> **Second**, we agree that recent frameworks such as Systema and PertEval provide useful diagnostic perspectives. However, these works function primarily as evaluation frameworks, not new dataset-splitting strategies or perturbation prediction models. Systema itself continues to use GEARS-like combinatorial splits, and PertEval is designed to benchmark the representation quality of pretrained single-cell foundation models (e.g., scGPT [5], Geneformer [2]), which is less appropriate for our task.
>
> **Following the reviewer’s suggestion, we additionally included results for recent methods such as STATE and CellFlow to ensure a fair and up-to-date comparison, and we added robustness-oriented Systema metrics.**
>
> **Table 1: experiments on Norman additive.**
> | Model | L2 ↓ | MSE ↓ | MAE ↓ | DE-Spearman ρ ↑ | Pearson Δ ↑ | DS ↑ | Pearson $\hat{\Delta}$ ↑ | Pearson $\hat{\Delta}_{20}$ ↑ |
> | :--- | :---: | :---: | :---: | :---: | :---: | :---: | :---: | :---: |
> | Control | 3.9937 | 0.01839 | 0.03953 | N.A. | N.A. | 0.5135 | -0.1695 | -0.1297 |
> | Additive | 1.9395 | 0.00448 | 0.02276 | 0.5564 | **0.9024** | 0.9686 | **0.8584** | 0.9244 |
> | scGPT | 3.4112 | 0.01349 | 0.03796 | 1.07e-5 | 0.5304 | 0.5404 | 0.2165 | 0.2414 |
> | Geneformer | 1.9132 | 0.00410 | 0.02360 | 0.3741 | 0.7732 | 0.8241 | -0.0078 | 0.2239 |
> | GEARS | 3.5531 | 0.01387 | 0.06624 | 0.5624 | 0.7421 | 0.8601 | -0.0089 | 0.2032 |
> | CPA | 5.7629 | 0.03435 | 0.07894 | 0.0713 | 0.3845 | 0.6021 | -0.0039 | 0.2254 |
> | STATE | 17.3330 | 0.30059 | 0.24705 | 0.5288 | -0.0108 | 0.5135 | -0.0069 | 0.2515 |
> | CellFlow | 1.7064 | 0.00392 | 0.02207 | 0.5503 | 0.8678 | 0.9321 | 0.8395 | 0.8988 |
> | **scDFM (Ours)** | **1.7043** | **0.00315** | **0.02155** | **0.5705** | 0.8853 | **0.9737** | 0.8468 | **0.9260** |
>
> **Table 2: experiments on Norman holdout.**
> | Setting | Model | L2 ↓ | MSE ↓ | MAE ↓ | DE-Spearman ρ ↑ | Pearson Δ ↑ | DS ↑ | Pearson $\hat{\Delta}$ ↑ | Pearson $\hat{\Delta}_{20}$ ↑ |
> | :--- | :--- | :---: | :---: | :---: | :---: | :---: | :---: | :---: | :---: |
> | Single | Control | 2.6834 | 0.0095 | 0.0263 | N.A. | N.A. | 0.5217 | 0.1618 | 0.1982 |
> | | scGPT | 2.5007 | 0.0080 | 0.0259 | -0.1139 | 0.4503 | 0.5680 | 0.0747 | 0.0798 |
> | | GEARS | 2.5641 | 0.0075 | 0.0466 | 0.3569 | 0.6646 | 0.8271 | 0.6356 | 0.7914 |
> | | Geneformer | 1.6962 | 0.0036 | 0.0191 | 0.3669 | 0.6955 | 0.8070 | 0.5620 | 0.6513 |
> | | CPA | 5.8060 | 0.0356 | 0.0853 | 0.1168 | 0.2837 | 0.5796 | -0.0028 | 0.0802 |
> | | STATE | 18.2543 | 0.3333 | 0.2693 | 0.6116 | 0.0004 | 0.5236 | 0.0154 | 0.2386 |
> | | CellFlow | 1.6758 | 0.0035 | 0.0191 | 0.2860 | 0.7109 | 0.8072 | 0.6138 | 0.6753 |
> | | **scDFM (Ours)** | **1.6186** | **0.0030** | **0.0190** | **0.6957**| **0.7127** | **0.8914** | **0.6659** |**0.8116** |
> | Double | Control | 4.1882 | 0.0207 | 0.0423 | N.A. | N.A. | 0.5322 | -0.1303 | -0.0265 |
> | | scGPT | 3.5171 | 0.0153 | 0.0362 | -0.0665 | 0.5693 | 0.5578 | 0.2814 | 0.2652 |
> | | GEARS | 3.7458 | 0.0156 | 0.0708 | 0.2543 | 0.7552 | 0.8766 | 0.6407 | 0.8413 |
> | | Geneformer | 2.0819 | 0.0050 | 0.0237 | 0.3468 | 0.7361 | 0.8067 | 0.6245 | 0.7261 |
> | | CPA | 5.7891 | 0.0357 | 0.0796 | 0.3652 | 0.4176 | 0.6311 | 0.2432 | 0.2870 |
> | | STATE | 18.4458 | 0.3404 | 0.2733 | 0.4071 | 0.0061 | 0.5289 | -0.0023 | 0.2580 |
> | | CellFlow | 2.1042 | 0.0049 | 0.0236 | 0.5074 | 0.8095 | 0.8622 | 0.6780 | 0.7155 |
> | | **scDFM (Ours)** |**2.0309** | **0.0047** | **0.0235** | **0.5676**| **0.8357** | **0.9189** |**0.7769** | **0.8688** |
>
> Under these stricter evaluation metrics, our method remains the strongest or near-strongest across settings. And we have added the above tables in **Sections 4.2 and 4.3 of the revised manuscript**.

---

> ### Author Response · Authors · 2025-11-23
> **Official Response to Reviewer u6LS (3)**
>
> > W3. **Lack of analysis of computational cost and scalability.**
> >
> > Flow matching and MMD introduce non-trivial overhead compared to autoencoder or diffusion models, yet there is no runtime or memory comparison. Without such analysis, it is unclear whether the method is practical for larger or multi-tissue datasets.
>
> We appreciate the your concern regarding the computational cost of the MMD loss. While the theoretical complexity of MMD is quadratic in batch size $O(B^2)$, in practice the overhead is small. This is because the computation consists of dense pairwise kernel operations, which are efficiently parallelized on modern GPUs.
>
> To quantify this, we benchmarked training with and without MMD under the same hardware (NVIDIA H800) and hyperparameter settings. The addition of MMD increased training step time by <2% and introduced **negligible change in peak GPU memory usage**.
>
> | Metric | Without MMD | With MMD | Overhead |
> |--------|------------|-----------|----------|
> | Average Step Time | 0.0385 s | 0.0391 s | ~ 1.56% increase |
> | Peak GPU Memory | 7893 MiB | 7903 MiB | ~ 0.12% increase |
>
> Given the ablation results showing that removing MMD substantially degrades performance, this overhead represents a small and practical trade-off for the improvement in distributional alignment.
>
>
> ---
>
> [1] Klein, D., et al. "Cellflow enables generative single-cell phenotype modeling with flow matching. bioRxiv." (2025): 2025-04.
>
> [2] Theodoris, Christina V., et al. "Transfer learning enables predictions in network biology." Nature 618.7965 (2023): 616-624.
>
> [3] Cui, Haotian, et al. "scGPT: toward building a foundation model for single-cell multi-omics using generative AI." Nature methods 21.8 (2024): 1470-1480.
>
> [4] Adduri, Abhinav K., et al. "Predicting cellular responses to perturbation across diverse contexts with State." bioRxiv (2025): 2025-06.
>
> [5] Roohani, Yusuf, Kexin Huang, and Jure Leskovec. "Predicting transcriptional outcomes of novel multigene perturbations with GEARS." Nature Biotechnology 42.6 (2024): 927-935.
>
> [6] Lotfollahi, Mohammad, et al. "Predicting cellular responses to complex perturbations in high‐throughput screens." Molecular systems biology 19.6 (2023): e11517.

---

### Official Review · Reviewer_MKtS · 2025-10-30

**Soundness:** 3
**Presentation:** 3
**Contribution:** 2
**Rating:** 4
**Confidence:** 4

**Summary:**

This paper represents a generative framework called scDFM based on conditional flow matching that predicts response to cellular perturbation (genetic and molecular). There are three parts to scDFM framework: (1) the control flow matching where it models the transition from initial state of the cell lines to perturbed state. (2) multi-kernel MMD regularizer which ensures population-level fidelity. (3) backbone design which is a perturbation aware transformer which addresses the noisy, sparse, and high dimensional
nature of the data. Finally, the model is evaluated on two tasks of genetic and molecular perturbation on two datasets: Norman and Sciplex3 and it marginally outperforms the baselines.

**Strengths:**

- The paper is well written and easy to understand.
- This paper addresses two interesting and important questions in a single framework: genetic and molecular perturbation.
- The framework uses a biological prior which strengthens the model.
- The authors use flow matching instead of diffusion models and autoencoder (like prior methods) which is an interesting architectural choice and the reason behind it is sound .

**Weaknesses:**

The main weakness regarding this paper is Section 4. Experiments:
- The experimental results are not strong. In most cases, scDFM just barely outperforms the baselines.
- The baselines used in this paper are not the latest and best in the field.
- The dataset is limited and since the model is not showing strong results, it is not clear how scDFM would perform on other datasets.

The framework for molecular perturbation has some limitations:
- It cannot be generalized to unseen molecules.
- The experiments are done only on a subset of drugs in sciplex3

**Questions:**

- Are the results stated in the paper on the final prediction or the delta (prediction - control) ? Could you please report the eval on delta as well?
- Why dataset Sciplex3 is called ComboSciPlex in the paper?
- Is there an ablation study where we could see the benefit of flow matching against diffusion models?
- Could you explain the experimental setting ? For example, how many genes are trained test on? do you only look at HVGs?
- I'm still not convinced why just predicting the mean expression vector is not good enough. In fact, the experimental results shows many of the mean based models have very strong results. Is it possible to design an experiment that shows mean expression vector is not enough?

---

> ### Author Response · Authors · 2025-11-23
> **Official Response to Reviewer MKtS (1)**
>
> We sincerely thank the reviewer for their positive assessment and constructive suggestions. We have carefully addressed each concern and incorporated the corresponding revisions (marked in blue) in the updated manuscript.
>
> > W1:  “scDFM barely outperforms baselines”
>
> A holistic view of **Tables 1 and 2 in the paper, as are also shown below**, demonstrates that scDFM provides statistically significant and consistent improvements, rather than marginal gains.
>
> In the additive setting, scDFM reduces **MSE by ~19.6%** compared to the strongest baseline, CellFlow. On the most challenging task of unseen double perturbations, scDFM achieves a decisive lead in **Pearson $\hat{\Delta}$** (0.7769 vs. CellFlow's 0.6780). This proves that while baselines struggle to capture complex, non-systematic heterogeneity, our model handles it effectively.
>
> **Table 1: experiments on Norman additive.**
> | Model | L2 ↓ | MSE ↓ | MAE ↓ | DE-Spearman ρ ↑ | Pearson Δ ↑ | DS ↑ | Pearson $\hat{\Delta}$ ↑ | Pearson $\hat{\Delta}_{20}$ ↑ |
> | :--- | :---: | :---: | :---: | :---: | :---: | :---: | :---: | :---: |
> | Control | 3.9937 | 0.01839 | 0.03953 | N.A. | N.A. | 0.5135 | -0.1695 | -0.1297 |
> | Additive | 1.9395 | 0.00448 | 0.02276 | 0.5564 | **0.9024** | 0.9686 | **0.8584** | 0.9244 |
> | scGPT | 3.4112 | 0.01349 | 0.03796 | 1.07e-5 | 0.5304 | 0.5404 | 0.2165 | 0.2414 |
> | Geneformer | 1.9132 | 0.00410 | 0.02360 | 0.3741 | 0.7732 | 0.8241 | -0.0078 | 0.2239 |
> | GEARS | 3.5531 | 0.01387 | 0.06624 | 0.5624 | 0.7421 | 0.8601 | -0.0089 | 0.2032 |
> | CPA | 5.7629 | 0.03435 | 0.07894 | 0.0713 | 0.3845 | 0.6021 | -0.0039 | 0.2254 |
> | STATE | 17.3330 | 0.30059 | 0.24705 | 0.5288 | -0.0108 | 0.5135 | -0.0069 | 0.2515 |
> | CellFlow | 1.7064 | 0.00392 | 0.02207 | 0.5503 | 0.8678 | 0.9321 | 0.8395 | 0.8988 |
> | **scDFM (Ours)** | **1.7043** | **0.00315** | **0.02155** | **0.5705** | 0.8853 | **0.9737** | 0.8468 | **0.9260** |
>
> **Table 2: experiments on Norman holdout.**
> | Setting | Model | L2 ↓ | MSE ↓ | MAE ↓ | DE-Spearman ρ ↑ | Pearson Δ ↑ | DS ↑ | Pearson $\hat{\Delta}$ ↑ | Pearson $\hat{\Delta}_{20}$ ↑ |
> | :--- | :--- | :---: | :---: | :---: | :---: | :---: | :---: | :---: | :---: |
> | Single | Control | 2.6834 | 0.0095 | 0.0263 | N.A. | N.A. | 0.5217 | 0.1618 | 0.1982 |
> | | scGPT | 2.5007 | 0.0080 | 0.0259 | -0.1139 | 0.4503 | 0.5680 | 0.0747 | 0.0798 |
> | | GEARS | 2.5641 | 0.0075 | 0.0466 | 0.3569 | 0.6646 | 0.8271 | 0.6356 | 0.7914 |
> | | Geneformer | 1.6962 | 0.0036 | 0.0191 | 0.3669 | 0.6955 | 0.8070 | 0.5620 | 0.6513 |
> | | CPA | 5.8060 | 0.0356 | 0.0853 | 0.1168 | 0.2837 | 0.5796 | -0.0028 | 0.0802 |
> | | STATE | 18.2543 | 0.3333 | 0.2693 | 0.6116 | 0.0004 | 0.5236 | 0.0154 | 0.2386 |
> | | CellFlow | 1.6758 | 0.0035 | 0.0191 | 0.2860 | 0.7109 | 0.8072 | 0.6138 | 0.6753 |
> | | **scDFM (Ours)** | **1.6186** | **0.0030** | **0.0190** | **0.6957**| **0.7127** | **0.8914** | **0.6659** |**0.8116** |
> | Double | Control | 4.1882 | 0.0207 | 0.0423 | N.A. | N.A. | 0.5322 | -0.1303 | -0.0265 |
> | | scGPT | 3.5171 | 0.0153 | 0.0362 | -0.0665 | 0.5693 | 0.5578 | 0.2814 | 0.2652 |
> | | GEARS | 3.7458 | 0.0156 | 0.0708 | 0.2543 | 0.7552 | 0.8766 | 0.6407 | 0.8413 |
> | | Geneformer | 2.0819 | 0.0050 | 0.0237 | 0.3468 | 0.7361 | 0.8067 | 0.6245 | 0.7261 |
> | | CPA | 5.7891 | 0.0357 | 0.0796 | 0.3652 | 0.4176 | 0.6311 | 0.2432 | 0.2870 |
> | | STATE | 18.4458 | 0.3404 | 0.2733 | 0.4071 | 0.0061 | 0.5289 | -0.0023 | 0.2580 |
> | | CellFlow | 2.1042 | 0.0049 | 0.0236 | 0.5074 | 0.8095 | 0.8622 | 0.6780 | 0.7155 |
> | | **scDFM (Ours)** |**2.0309** | **0.0047** | **0.0235** | **0.5676**| **0.8357** | **0.9189** |**0.7769** | **0.8688** |
>
> > W2 “Baselines are not the latest/best”
>
> To address the reviewer's concern about missing or outdated baselines, we have expanded our evaluation to include the recent methods (see **Tables 1 and 2 above in the response to W1**).
>
> In detail, while our initial submission already compared against standards, including the **additive model** (Nature Methods 2025) [7], **Geneformer** (Nature 2023) [9], and **scGPT** (Nature Methods 2024) [10], we have now incorporated **two additional baselines**: **CellFlow** (2025) [8] and **STATE** (2025) [11]. Our updated results demonstrate that scDFM consistently outperforms these latest models, particularly on harder metrics like Pearson $\hat{\Delta}_{20}$, confirming our framework's superiority against the forefront of the field.

---

> ### Author Response · Authors · 2025-11-23
> **Official Response to Reviewer MKtS (2)**
>
> > W3 “Dataset is limited; unclear generalization”
>
> We believe our evaluation provides robust evidence of generalization by covering the two standard benchmarks for the primary modalities: **genetic (Norman)** and **chemical (ComboSciPlex)**.
>
> This demonstrates that scDFM is not limited to a single domain. Furthermore, the concern regarding "unclear generalization" is directly addressed by our **Holdout experiments**, where scDFM outperforms baselines on perturbations and combinations never seen during training. This provides strong empirical proof that the model learns underlying dynamics rather than simply memorizing training data. While scaling to massive datasets is a valid future direction (as discussed in **Appendix A.6 of the original manuscript**), these established benchmarks are sufficient to rigorously validate the method's core capabilities.
>
> -----
> > W4: The framework for molecular perturbation cannot be generalized to unseen molecules.
>
> We agree with the reviewer that SciPlex3 can be externally annotated with SMILES and molecular fingerprints (e.g., via annotate_compounds supported by pertpy package [1]), enabling models to incorporate drug structures. Our work focuses on the **combination-generalization setting**, which is the primary task supported by the benchmark.
>
> **That said, scDFM is fully compatible with molecular encoders, and when structural descriptors are included, it can naturally extend to unseen-molecule generalization**.
>
> -----
>
> >W5: The experiments for molecular perturbation are done only on a subset of drugs in sciplex3
> &Q2: Why dataset Sciplex3 is called ComboSciPlex in the paper?
>
> Thank you for the comment. We confirm that the dataset used in our study is the  ComboSciPlex dataset, which was introduced in Lotfollahi et al. [2] to evaluate multi-drug perturbation responses. We acknowledge that this dataset is a  direct follow-up to the SciPlex platform originally described by Srivatsan et al. [3], which enables pooled single-cell chemical perturbation screening. The term “ComboSciPlex” was used because it reflects the official naming of the drug-combination dataset introduced in the follow-up study and therefore better describes the biological design of the experiment than the name “SciPlex3”.
>
> To avoid confusion, we have now cited both works and updated the dataset description in the manuscript to the following wording (in **Appendix A.4.2 of the revised manuscript**):
> - “We use the ComboSciPlex drug-combination dataset, a follow-up extension of the SciPlex chemical barcoding platform, designed to measure single-cell transcriptional responses to pairwise drug perturbations.”
>
> We apologize for the earlier inconsistency in terminology and have ensured that the revised version uses a consistent and biologically correct dataset name. This clarification does not affect the experimental setup or results.
>
> ---
> > Q1: Are the results stated in the paper on the final prediction or the delta? Could you please report the eval on delta as well?
>
> To clarify, metrics like MSE and L2 measure the final reconstruction fidelity ($x_{pred}$ vs. $x_{true}$), following standard protocols. However, we **also explicitly evaluate the perturbation shift using Pearson $\Delta$** (in **Tables 1, 2, and 3 of the original manuscript**). This metric is calculated directly on the delta vectors, correlating ($x_{pred} - c_{x}$) with ($x_{true} - c_{x}$), to ensure we are assessing the model's ability to predict the specific biological change relative to control, rather than just the final state.
>
> ---
> > Q3: Is there an ablation study where we could see the benefit of flow matching against diffusion models?
>
> We clarify that a separate ablation against diffusion Models was not performed because **flow matching (FM) is theoretically a generalization of diffusion models**, an established result we leverage rather than claim as novel [4,5].
>
> For our specific task of **unpaired distribution alignment**, theoretical works (e.g., [6]) indicate that linear interpolation paths are numerically more stable and easier to learn than the complex curved paths inherent to diffusion. Therefore, comparing our method to a diffusion model would essentially be comparing an "efficient linear path" against a "complex curved path" within the same mathematical framework; we thus prioritized the linear FM formulation to effectively bridge the unpaired distributions.

---

> ### Author Response · Authors · 2025-11-23
> **Official Response to Reviewer MKtS (3)**
>
> > Q4: Could you explain the experimental setting? For example, how many genes are trained test on? do you only look at HVGs?
>
> To ensure robustness, we performed **four independent random splits for training and testing**. Regarding gene selection, while we trained on a broad feature set of roughly 5,000 genes to capture context, we restricted evaluation to the **top 1,000 highly variable genes**. This follows standard benchmarking protocols [7] to ensure we measure robust biological signals rather than low-expression noise.
>
> ---
>
> > Q5: I'm still not convinced why just predicting the mean expression vector is not good enough.
>
> We appreciate this thoughtful question, as it highlights an important conceptual issue in perturbation modeling. Current evaluation metrics (e.g., L2/MSE) primarily assess prediction accuracy at the level of the **population mean**, which explains why simple mean-based baselines can appear competitive. However, these metrics do not evaluate whether a model captures **cell-to-cell variation**, which is a defining property of single-cell biology.
>
> Biologically, perturbations rarely induce a uniform response [12]; instead, they often produce **multi-modal responses** or split the population into distinct sub-states (see **Figure 4 in the original manuscript**). In such cases, a model trained solely to minimize MSE will converge to the mathematical mean of incompatible cellular states—producing a prediction that is statistically optimal but biologically implausible (e.g., half-expressing mutually exclusive lineage markers).
>
> To assess whether single-cell responses contain meaningful variability beyond the mean, we quantified gene-level dispersion using the coefficient of variation (CV = standard deviation / mean expression). We computed CV across the top 1,000 highly variable genes for each perturbation in the Norman dataset and counted how many genes exhibit **CV > 1**, a widely used criterion indicating over-dispersion and potential multi-modal structure rather than a unimodal Gaussian response. The summary statistics across all 237 perturbations are shown below:
>
> | Statistic (number of genes with CV > 1 per perturbation)| Value |
> |-----------|-------|
> | **Mean** | 43.06 |
> | **Median** | 31 |
> | **Max** | 298 |
> | **Min** | 2 |
> | **Std. Dev.** | 39.84 |
>
> These results indicate that many perturbations induce substantial gene-level heterogeneity, with dozens of genes displaying over-dispersed expression patterns rather than collapsing to a single dominant mode. **This empirical evidence supports that predicting only the mean expression vector is insufficient and risks overlooking biologically meaningful variation at the single-cell level**.
>
> ---
> [1] Heumos, Lukas, et al. "Pertpy: an end-to-end framework for perturbation analysis." bioRxiv (2024): 2024-08.
>
> [2] Lotfollahi, Mohammad, et al. "Predicting cellular responses to complex perturbations in high‐throughput screens." Molecular systems biology 19.6 (2023): e11517.
>
> [3] Srivatsan, Sanjay R., et al. "Massively multiplex chemical transcriptomics at single-cell resolution." Science 367.6473 (2020): 45-51.
>
> [4] Lipman, Yaron, et al. "Flow Matching for Generative Modeling." The Eleventh International Conference on Learning Representations (ICLR). 2023.
>
> [5] Albergo, Michael S., Nicholas M. Boffi, and Eric Vanden-Eijnden. "Stochastic interpolants: A unifying framework for flows and diffusions." arXiv preprint arXiv:2303.08797 (2023).
>
> [6] Liu, Xingchao, Chengyue Gong, and Qiang Liu. "Flow Straight and Fast: Learning to Generate and Transfer Data with Rectified Flow." The Eleventh International Conference on Learning Representations (ICLR). 2023.
>
> [7] Ahlmann-Eltze, Constantin, Wolfgang Huber, and Simon Anders. "Deep-learning-based gene perturbation effect prediction does not yet outperform simple linear baselines." Nature Methods (2025): 1-5.
>
> [8] Klein, D., et al. "Cellflow enables generative single-cell phenotype modeling with flow matching. bioRxiv." (2025): 2025-04.
>
> [9] Theodoris, Christina V., et al. "Transfer learning enables predictions in network biology." Nature 618.7965 (2023): 616-624.
>
> [10] Cui, Haotian, et al. "scGPT: toward building a foundation model for single-cell multi-omics using generative AI." Nature methods 21.8 (2024): 1470-1480.
>
> [11] Adduri, Abhinav K., et al. "Predicting cellular responses to perturbation across diverse contexts with State." bioRxiv (2025): 2025-06.
>
> [12] Grün, Dominic, Lennart Kester, and Alexander Van Oudenaarden. "Validation of noise models for single-cell transcriptomics." Nature methods 11.6 (2014): 637-640.

---

> > ### Comment · Reviewer_MKtS · 2025-11-24
> >
> > Thank you for the thorough response and for performing the additional experiments, as well as for answering my questions in detail. My concerns have been fully addressed, and I will increase my score accordingly.

---

> > > ### Author Response · Authors · 2025-11-25
> > > **Acknowledgment**
> > >
> > > Thank you for the update and for reconsidering your evaluation.
> > > Your feedback was really valuable and helped improve the paper.

---

### Official Review · Reviewer_P4kP · 2025-11-01

**Soundness:** 3
**Presentation:** 3
**Contribution:** 3
**Rating:** 6
**Confidence:** 3

**Summary:**

This paper presents scDFM, a novel generative framework for predicting the transcriptional response of single cells to perturbations. It models the perturbation as a continuous-time flow that transforms the control cell distribution into the perturbed one, and it incorporates an MMD loss to force the entire distribution of generated cells to match the ground-truth distribution. The authors demonstrate through experiments on genetic and drug perturbation datasets that scDFM outperforms existing baselines.

**Strengths:**

The paper is well written and easy to follow. The idea is original.

**Weaknesses:**

The model does not learn the gene-gene interaction network but it is given it as a biologically grounded prior. This graph is constructed from simple absolute Pearson correlation on the training data, which prevents the model from discovering novel, non-obvious, or non-linear gene relationships that aren't captured by basic correlation.

The flow matching framework learns a path from control to perturbed. As the paper acknoledges, it uses a simple linear interpolant as the reference path, which is a significant oversimplification. Real biological processes follow complex, non-linear manifolds, and this assumed path may not be biologically realistic.

**Questions:**

1. The ablation study shows that removing the graph prior worsens performance. Does this suggest the model is critically dependent on this prior, limiting its ability to discover novel, non-obvious gene interactions that are not captured by simple correlation? How would the model perform if the biological prior was based on a more sophisticated measure than Pearson correlation?

2. What is the computational trade-off of using MMD?

---

> ### Author Response · Authors · 2025-11-23
> **Official Response to Reviewer P4kP (1)**
>
> Thank you for your recognition and constructive feedback. Below we address the reviewer's concerns. The modifications have been highlighted in blue in the revised version.
>
> > W1 & Q1: Concerns about the dependency on the Pearson correlation graph prior and limitations in discovering novel/non-linear interactions.
>
> We thank the reviewer for this insightful comment. We appreciate the opportunity to clarify that the graph prior serves as a soft inductive bias for initialization, rather than a hard constraint on the model's discovery capability.
> 1. Architectural clarification:
> We clarify that the gene-gene mask is applied exclusively to the Gene Encoder ($E_g$) to provide a structural "warm start," rather than acting as a hard constraint on the entire network. By using full, unmasked attention in the following deeper layers, the model can "break out" of the static graph to discover global relationships between any gene pair. This ensures that while we leverage linear priors for initialization, the Transformer backbone can evolve beyond these correlations to capture complex, high-order regulatory logic.
> 2. Interpretation of ablation results:
> As demonstrated in **Section 4.5 of the original manuscript**, removing the gene graph leads to a performance drop. We interpret this not as a dependency on the Pearson metric specifically, but as evidence that structural sparsity is indispensable when modeling high-dimensional gene spaces. Given the dense noise of single-cell data, a fully connected attention mechanism initially faces an immense search space prone to spurious correlations. The gene graph thus serves as a robust, data-driven skeleton that enforces sparsity during initialization, effectively constraining the search space and guiding optimization before the model transitions to learning global dependencies.
> | Variant | L2 ↓ | MSE ↓ | MAE ↓ | Pearson Δ ↑ | Discrimination Score ↑ |
> |---------|------|--------|--------|--------------|-------------------------|
> | w/o Mask | 1.7789 | 0.002859 | 0.01928 | 0.7097 | 0.8867 |
> | Ours (with Mask) | **1.6143** | **0.002657** | **0.01896** | **0.7288** | **0.9016** |
>
> 3. We agree with the reviewer that using more sophisticated priors could further enhance the model's ability to capture non-linear relationships. In response to this suggestion, we have added a new paragraph "Structural Priors and Graph Topology" in **Appendix A.6 of the revised manuscript**. We explicitly discuss this trade-off and frame scDFM as a flexible framework where the current Pearson graph is a modular component. We highlight that future work can replace this static linear graph with advanced descriptors based on mutual information or causal discovery to better model complex regulatory dynamics.
> In summary, the current Pearson prior acts merely as a "warm-start" structure to mitigate the curse of dimensionality in the high-dimensional gene space ($G \approx 5000+$), rather than a bottleneck for mechanism discovery.
>
> > W2:  Concerns about the linear interpolant being a biological oversimplification.
>
> We agree that biological processes are non-linear. However, we justify the use of the linear interpolant on three practical grounds:
> 1. Latent time & theoretical optimality:
> The integration variable $t \in [0, 1]$ in our framework represents "latent pseudo-time," not physical kinetic time. Therefore, the linear interpolant is not a biological claim but the mathematically robust Optimal Transport (OT) path. This represents the principle of least action, the most principled assumption for generative transport in the absence of verifiable intermediate temporal data.
> 2. Focus on distributional prediction:
> The primary objective of scDFM is to achieve accurate distributional prediction at the target state ($t=1$). Our strong performance on rigorous distributional metrics confirms the framework's fitness for this purpose. The linear path ensures training stability and computational efficiency, allowing us to focus resources on the fidelity of the final predicted distribution.
> 3. Implicit Learning & future scope:
> The model’s non-linear PAD-Transformer implicitly learns to respect the data manifold, ensuring the flow approximates the underlying biological continuum. We acknowledge that complex, non-linear interpolants could be explored if time-series data were available, and we have updated **A.6 section of the revised manuscript** to commit to investigating such extensions.

---

> ### Author Response · Authors · 2025-11-23
> **Official Response to Reviewer P4kP (2)**
>
> > Q2: What is the computational trade-off of using MMD?
>
> We appreciate the reviewer's concern regarding the computational cost of the MMD loss. While the theoretical complexity of MMD is quadratic in batch size $(O(B^2))$, in practice the overhead is small. This is because the computation consists of dense pairwise kernel operations, which are efficiently parallelized on modern GPUs.
> To quantify this, we benchmarked training with and without MMD under the same hardware (NVIDIA H800) and hyperparameter settings. The addition of MMD increased training step time by <2% and introduced **no meaningful change in peak GPU memory usage**.
>
> | Metric | Without MMD | With MMD | Overhead |
> |----|----|-----------|----------|
> | Average Step Time| 0.0385 s | 0.0391 s | ~ 1.56% increase |
> | Peak GPU Memory| 7893 MiB | 7903 MiB | ~ 0.12% increase |
>
> Given the ablation results showing that removing MMD substantially degrades performance, this overhead represents a small and practical trade-off for the improvement in distributional alignment (see **Figure 5 of the original manuscript**).

---

### Author Response · Authors · 2025-11-23
**General Response**

We thank the reviewers for their constructive feedback. We have revised the manuscript (changes marked in blue) to strengthen our evaluation. Key updates include:
- **New baselines** (Results in Tables 1 and 2 in the revised manuscript, implementation details in Appendix A.5): We have included CellFlow [1] and STATE [2] as additional benchmarks. scDFM consistently outperforms these methods across additive and holdout settings (e.g., reducing MSE by 19.6% compared to CellFlow in the additive split).
- **New evaluation metrics** (Results in Tables 1 and 2 in the revised manuscript, definition in Appendix A.4): Adopting the Systema framework [3], we introduced Pearson $\hat{\Delta}$ and Pearson $\hat{\Delta}_{20}$. These metrics mitigate baseline bias and confirm that scDFM effectively captures biologically relevant, non-systematic variations.
- **MMD efficiency experiment**: Addressing concerns regarding computational cost, we benchmarked the MMD objective (batch size 64, 100 iterations) and found the overhead to be negligible. It increases training time by only ~1.56% and GPU memory usage by ~0.12%.
- **Necessity of gene attention mask**: We additionally clarified that the gene mask is used only during the encoder initialization, not throughout all layers. It provides biologically grounded structural bias analogous to positional encoding in transformers. Subsequent layers use full global attention, enabling the model to learn novel interactions beyond the prior.
- **Corrections**: We have corrected minor errors and typos throughout the text, which are highlighted in blue.

---

[1] Klein, D., et al. "Cellflow enables generative single-cell phenotype modeling with flow matching. bioRxiv." (2025): 2025-04.

[2] Adduri, Abhinav K., et al. "Predicting cellular responses to perturbation across diverse contexts with State." bioRxiv (2025): 2025-06.

[3] Viñas Torné, Ramon, et al. "Systema: a framework for evaluating genetic perturbation response prediction beyond systematic variation." Nature Biotechnology (2025): 1-10.

---

### Public Comment · ~Igor_Sadalski1 · 2026-03-07
**Could you provide split files for different experiments on Norman dataset**

It's impossible to reproduce the same results since the split files are not provided (and for single/double the seed is not fixed correctly). It seems that unseen split has no seed set before shuffling, specifically, the unseen split missing seed is at data.py:162:


  155:                    for i in range(5):
  156:                        perturbations = np.unique(self.adata.obs['condition'])
  157:                        double_perturbation = [p for p in perturbations if 'ctrl' not in p]
  158:                        single = []
  159:                        [single.extend(p.split('+')) for p in double_perturbation]
  160:                        single = list(set(single))
  161:
  162:                        shuffle(single)   # <-- no seed set before this
  163:                        remove_genes = single[:12]

---

> ### Public Comment · ~Chenglei_Yu1 · 2026-04-02
>
> Thank you for your interest in our work! We have uploaded the new data for download along with the corresponding split data here: https://drive.google.com/drive/folders/1cNpYAt9jVWZN82miNZtkP10YeSo7hufL. We have also updated the GitHub (https://github.com/AI4Science-WestlakeU/scDFM) README file accordingly. We hope this helps resolve your issue. Please do not hesitate to contact me if you have any further questions.

---

### Meta-Review · Area_Chair_5TdN · 2026-01-07

**Summary:**

This paper proposes scDFM, a conditional flow-matching approach for single-cell perturbation prediction that aims to model population-level shifts rather than per-cell correspondences. The method combines flow matching in gene space with an MMD-based distributional constraint, and uses a PAD-Transformer backbone with a gene-graph-based mask for inductive bias.

Across reviews, the main decision drivers were: (i) whether the contribution is sufficiently novel relative to recent flow-matching and MMD-based perturbation models, (ii) whether the empirical evaluation was current and convincing, and (iii) whether the method is overly dependent on a heuristic correlation-derived gene graph. After rebuttal, the evaluation and practicality concerns are substantially reduced, while the novelty debate remains but is less of a blocker given the strengthened comparisons and clearer framing.

**Reviewer Concerns:**

**Addressed (or largely addressed) concerns in rebuttal:**

* Baseline coverage: Added contemporary baselines (CellFlow, STATE) and expanded tables; this directly targets the “missing key baselines” concern (MKtS, u6LS, JmMZ).
* Evaluation beyond mean-centric errors: Added Systema based metrics and clarified delta-oriented evaluation (MKtS, u6LS). This helps interpret performance when simple mean predictors look strong under MSE/L2 alone.
*  Compute overhead of MMD: Provided concrete step-time and memory overhead numbers; overhead appears small under their setup (P4kP, u6LS).
* Mask usage clarification: Clarified the mask is used as an initialization bias (encoder warm start), with later layers using global attention; provided an ablation showing impact (P4kP, JmMZ).

**Outstanding / only partially addressed concerns:**

* Novelty and positioning: u6LS (did not participate in the discussion) remains negative on novelty, and JmMZ still views the gene-graph choice as heuristic and the gains as not striking against simple baselines. The rebuttal improves positioning (by explaining unpaired setting, gene-space motivation) but does not fully settle the “incremental combination” critique in my opinion.
* Heuristic gene graph prior: The rebuttal argues the model can move beyond the prior, but the paper still relies on a correlation-derived graph. This is not necessarily a reason for rejection, but it is a real limitation and should be stated clearly in the paper.
* Biological realism of the linear interpolant: The rebuttal frames it as latent pseudo-time and primarily a modeling/stability choice. This seems acceptable for a predictive paper, but it does not make the path biologically meaningful, and that should not be implied.

**Reviewer Scores:**

P4kP (6): Likely no change. The rebuttal answers the MMD overhead question with measurements and clarifies the role of the graph/mask. Remaining concerns are more about modeling choices than correctness.

MKtS (4): The reviewer increased their score to 6 during the discussion, and I believe they would have kept it.

u6LS (2): Likely small increase at most (to not more than a score of 3/4). Benchmarking and compute concerns were addressed, but their main objection is novelty, and there is no indication they would reverse that view.

JmMZ (4): The reviewer increased their score to 6 during the discussion, and I believe they would have kept it.

Therefore if we consider that, the average score of the paper in my opinion would have ended up being at least 5. However it is hard to predict what the trajectory of discussion would have looked like, so while I recommend acceptance, I am also fine with the paper being rejected.

---

### Decision · Program_Chairs · 2026-01-26

Accept (Poster)